# GemNet: Universal Directional Graph Neural Networks for Molecules

**Johannes Gasteiger, Florian Becker, Stephan Günnemann**
Technical University of Munich, Germany
`{j.gasteiger,beckerf,guennemann}@in.tum.de`

## Abstract

Effectively predicting molecular interactions has the potential to accelerate molecular dynamics by multiple orders of magnitude and thus revolutionize chemical simulations. Graph neural networks (GNNs) have recently shown great successes for this task, overtaking classical methods based on fixed molecular kernels. However, they still appear very limited from a theoretical perspective, since regular GNNs cannot distinguish certain types of graphs. In this work we close this gap between theory and practice. We show that GNNs with directed edge embeddings and two-hop message passing are indeed universal approximators for predictions that are invariant to translation, and equivariant to permutation and rotation. We then leverage these insights and multiple structural improvements to propose the geometric message passing neural network (GemNet). We demonstrate the benefits of the proposed changes in multiple ablation studies. GemNet outperforms previous models on the COLL, MD17, and OC20 datasets by 34 %, 41 %, and 20 %, respectively, and performs especially well on the most challenging molecules. Our implementation is available online. [1]

## 1 Introduction

Graph neural networks (GNNs) have shown great promise for predicting the energy and other quantum mechanical properties of molecules. They can predict these properties orders of magnitudes faster than methods from quantum chemistry – at comparable accuracy. GNNs can thus enable the accurate simulation of systems that are orders of magnitude larger. However, they still exhibit severe theoretical and practical limitations. Regular GNNs are only as powerful as the 1-Weisfeiler Lehman test of isomorphism and thus cannot distinguish between certain molecules [45, 60]. Moreover, they require a large number of training samples to achieve good accuracy.

In this work we first resolve the questionable expressiveness of GNNs by proving sufficient conditions for universality in the case of invariance to translations and rotations and equivariance to permutations; and then extending this result to rotationally equivariant predictions. Simply using the full geometric information (e.g. all pairwise atomic distances) in a layer does not ensure universal approximation. For example, if our model uses a rotationally invariant layer we lose the relative information between components. Such a model thus cannot distinguish between features that are rotated differently. This issue is commonly known as the "Picasso problem": An image model with rotationally invariant layers cannot detect whether a person's eyes are rotated correctly. Instead, we need a model that preserves relative rotational information and is only invariant to *global* rotations. To prove universality in the rotationally invariant case we extend a recent universality result based on point cloud models that use representations of the rotation group $SO(3)$ [18]. We prove that spherical representations are actually sufficient; full $SO(3)$ representations are not necessary. We then generalize this to rotationally equivariant predictions by leveraging a recent result on extending invariant to equivariant

---

[1] `https://www.daml.in.tum.de/gemnet`

35th Conference on Neural Information Processing Systems (NeurIPS 2021).

predictions [57]. We then discretize spherical representations by selecting points on the sphere based on the directions to neighboring atoms. We can connect this model to GNNs by interpreting these directions as directed edge embeddings. For example, the embedding direction of atom $a$ would be defined by atom $c$, resulting in the edge embedding $e_{ca}$. Updating the spherical representation of atom $a$ based on atom $b$ then corresponds to two-hop message passing between the edges $e_{ca}$ and $e_{db}$ via $e_{ba}$, with atoms $c$ and $d$ defining the embedding directions. This message passing formalism naturally allows us to obtain the molecule's full geometrical information (distances, angles, and dihedral angles), and the direct correspondence proves the model's universality.

We call this edge-based two-hop message passing scheme *geometric message passing*, and propose multiple structural enhancements to improve the practical performance of this formalism. Based on these changes we develop the highly accurate and sample-efficient geometric message passing neural network (GemNet). We furthermore show that stabilizing the variance of GemNet's activations with predetermined scaling factors yields significant improvements over regular normalization layers.

We investigate the proposed improvements in a range of ablation studies, and show that each of them significantly reduces the model error. These changes introduce little to no computational overhead over two-hop message passing. Altogether, GemNet outperforms previous models for force predictions on COLL by 34 %, on MD17 by 41 %, and on OC20 by 20 % on average. We observe the largest improvements for the most challenging molecules, which exhibit dynamic, non-planar geometries. In summary, our contributions are:

- Showing the **universality** of spherical representations and two-hop message passing with directed edge embeddings for rotationally equivariant predictions.
- **Geometric message passing**: Symmetric message passing enhanced by geometric information.
- Incorporating all proposed improvements in the **Geometric Message Passing Neural Network (GemNet)**, which significantly outperforms previous methods for molecular dynamics prediction.

## 2 Related work

**Machine learning potentials.** Research on predicting a molecule's energy and forces (so-called machine learning potentials) started with hand-fitted analytical functions and then gradually moved towards fully learned models. Arguably, classical force fields are their very first instances. They use analytical functions with coefficients that were hand-tuned based on experimental data. A popular example for these is the Merck Molecular Force Field (MMFF94) [31]. The next wave of methods used kernel ridge regression based on fixed, hand-crafted molecular representations [3, 9, 23]. Finally, modern research mostly focusses on fully end-to-end learnable models based on GNNs [30, 53]. These models can also be combined with molecular features from quantum mechanical calculations to improve performance [48]. We consider this combination as orthogonal research.

**Directional GNNs.** We can also achieve equivariance and invariance to rotations without relying on group representations. Directional GNNs achieve this by representing directional information explicitly [54] or in the form of angles [29] and dihedral angles [25, 39]. Our work is focused on this class of models, proving their universality and proposing an improved variant, GemNet.

**Expressiveness of GNNs.** A large part of GNN research has been focused on their (limited) expressiveness. Morris et al. [45], Xu et al. [60] first proved that they are only as expressive as the Weisfeiler-Lehman test of isomorphism and Garg et al. [27] showed the limitations of basic directional message passing. Kondor et al. [37], Maron et al. [43], Morris et al. [45, 46] then investigated higher-order representations to circumvent this issue. Finally, Azizian & Lelarge [2], Maron et al. [42] showed that so-called folklore GNNs are the most expressive GNNs for a given tensor order.

**Equivariant neural networks.** Equivariance and invariance have recently emerged as one of the foundational principles of modern neural networks [13, 15]. This is especially relevant for models in physics, for which we often know the symmetries a priori. Equivariant models for the $SO(3)$ group were first investigated in the context of spherical convolutions by Cohen et al. [14], Esteves et al. [21], Kondor et al. [36]. These methods leverage group representations to achieve full equivariance. They were then transferred to the context of 3D point clouds and molecules by Anderson et al. [1], Thomas et al. [56], Weiler et al. [58], and further developed by Batzner et al. [4], Finzi et al. [24], Fuchs et al. [26]. Importantly, Yarotsky [61] proved the universality of 2D convolutional networks, and Bogatskiy et al. [5] extended this result to general groups. Maron et al. [44] proved

universality for models invariant to $S_n$ and equivariant to an additional symmetry. Dym & Maron [18] combined these results to prove universality for the joined group of translations, rotations, and permutations. Apart from reflections this is the exact group relevant for general molecules.

## 3 Universality of spherical representations

GNNs for molecules typically incorporate directional information in one of two ways: Via $SO(3)$ representations [1, 56] or by using directions in real space [29, 54]. Directions in real space are associated with the three-dimensional $S^2$ sphere, while the $SO(3)$ group is double covered by the four-dimensional $S^3$ sphere. Directional representations thus use one degree of freedom less than $SO(3)$ representations, making them significantly cheaper. And, as we will prove in this section, directional representations actually provide the same expressivity as $SO(3)$ representations for predictions in $\mathbb{R}^3$. We achieve this by showing that the $SO(3)$-based tensor field network (TFN) [56] variant used by Dym & Maron [18] is equivalent to a similar model based on spherical representations, in the case of rotationally invariant predictions. We then generalize a recent result by Villar et al. [57], which lets us extend our theorem to the rotationally equivariant case. Afterwards, we relate this universality to directional GNNs by interpreting them as a discretization of spherical representations.

**Preliminaries.** We consider a point cloud with $n$ points (atoms), each associated with a position and a set of rotationally invariant features (e.g. atom types), defined as $\boldsymbol{X} \in \mathbb{R}^{3 \times n}$ and $\boldsymbol{H}_{\text{in}} \in \mathbb{R}^{h \times n}$. In this section we define model classes by sets of functions $\mathcal{F}$. As a first step, we are interested in proving that the set $\mathcal{F}$ defining our model is equal to the full set of functions $\mathcal{G}'$ that are invariant to the group of translations $\mathbb{T}^3$ and rotations $SO(3)$, and equivariant to the group of permutations $S_n$. We denote the codomain of functions in $\mathcal{G}'$ as $W_{\text{T}}^n$, where $W_{\text{T}}$ is some representation of $SO(3)$. We denote a vector's norm by $x = \|\boldsymbol{x}\|_2$, its direction by $\hat{\boldsymbol{x}} = \boldsymbol{x}/x$, and the relative position by $\boldsymbol{x}_{ba} = \boldsymbol{x}_b - \boldsymbol{x}_a$. Proofs are deferred to the appendix. Note that this section is not intended as an introduction to the $SO(3)$ group. For a concise introduction in the context of machine learning see e.g. Weiler et al. [58, Section 3] or Kondor et al. [36].

**Tensor field network.** In order to show the equivalence of the TFN to spherical representations, we first need to define this model. Following Dym & Maron [18], we split the model into two parts: Embedding functions $\mathcal{F}_{\text{feat}}$ that lifts the input into an equivariant representation, and pooling functions $\mathcal{F}_{\text{pool}}$ that aggregate the results of multiple embedding functions on each point and computes the model output. The overall model is then defined as the set of functions

$$\mathcal{F}_{K(D),D}^{\text{TFN}} = \{f \mid f(\boldsymbol{X}, \boldsymbol{H}_{\text{in}}) = \sum_{k=1}^{K} f_{\text{pool}}^{(k)*}(f_{\text{feat}}^{(k)}(\boldsymbol{X}, \boldsymbol{H}_{\text{in}})), f_{\text{pool}}^{(k)} \in \mathcal{F}_{\text{pool}}^{\text{TFN}}(D), f_{\text{feat}}^{(k)} \in \mathcal{F}_{\text{feat}}^{\text{TFN}}(D)\},$$

(1)

where $D \in \mathbb{N}$ denotes the function's maximum polynomial degree, $K(D) \in \mathbb{N}$ is chosen such that Theorem 1 is fulfilled (Dym & Maron [18] only prove the existence of this function), and $f^*$ denotes elementwise application of $f$ on all points. We then define the set $\mathcal{F}_{\text{pool}}^{\text{TFN}}$ as all rotationally equivariant linear functions on the $SO(3)$ group, i.e. all $SO(3)$ convolutions [38]. Note that these are more expressive than the self-interaction layers used originally [56]. The embedding functions $\mathcal{F}_{\text{feat}}^{\text{TFN}}(D) = \{\pi_2 \circ f^{(2D)} \circ \cdots \circ f^{(1)} \mid f^{(i)} \in \mathcal{F}_{\text{prod}}^{\text{TFN}}\}$ consist of an auxiliary function $\pi_2(\boldsymbol{X}, \boldsymbol{H}) = \boldsymbol{H}$ and a series of tensor product functions (called convolution by Dym & Maron [18]) $\mathcal{F}_{\text{prod}}^{\text{TFN}} = \{f \mid f(\boldsymbol{X}, \boldsymbol{H}) = (\boldsymbol{X}, \tilde{\boldsymbol{H}}^{\text{TFN}}(\boldsymbol{X}, \boldsymbol{H}))\}$. The intermediate representations are $\boldsymbol{H} \in W_{\text{feat}}^n$, where $W_{\text{feat}}$ is a representation of $SO(3)$ indexed by the degree $l$ and the order $m$. For $\boldsymbol{H}_{\text{in}}$ we have $l = m = 0$. The main update is defined by

$$\tilde{\boldsymbol{H}}_{am_o}^{\text{TFN}(l_o)}(\boldsymbol{X}, \boldsymbol{H}) = \theta \boldsymbol{H}_{am_o}^{(l_o)} + \sum_{l_f, m_f} \sum_{l_i, m_i} C_{(l_f, m_f),(l_i, m_i)}^{(l_o, m_o)} \sum_{b \in \mathcal{N}_a} F_{\text{TFN}, m_f}^{(l_f)}(\boldsymbol{x}_b - \boldsymbol{x}_a) \boldsymbol{H}_{bm_i}^{(l_i)}, \quad (2)$$

where $\theta$ is a (learned) scalar and $\mathcal{N}_a$ are the neighbors of point $a$. The Clebsch-Gordan coefficients $C_{(l_f, m_f),(l_i, m_i)}^{(l_o, m_o)}$ arise from decomposing the tensor product of two input $SO(3)$ representations (the filter and input representations) into a sum of output representations. Their exact values are not relevant for this discussion. We index the output with degree $l_o$ and order $m_o$, the learned filter with $l_f$ and $m_f$, and the input with $l_i$ and $m_i$. $F_{\text{TFN}, m}^{(l)}(\boldsymbol{x}) = R^{(l)}(x) Y_{lm}(\hat{\boldsymbol{x}})$ is a rotationally equivariant

filter, with a (learned) radial part $R$, which is any polynomial of degree $\leq D$, and the real spherical harmonics $Y_{lm}$ with degree $l$ and order $m$. The spherical harmonics are the basis for the Fourier transformation of functions on the sphere, analogously to sine waves for functions on $\mathbb{R}$. We can prove universality for TFNs by using the universality of polynomial regression and showing that TFNs can fit any polynomial (see Dym & Maron [18] for details), resulting in:

**Theorem 1** (Dym & Maron [18]). *Consider the set of functions $\mathcal{G}$ mapping $\mathbb{R}^{3 \times n + h \times n} \to W_{\mathrm{T}}^n$ that are equivariant to rotations and permutations and invariant to translations. For all $n \in \mathbb{N}$,*

1. *For $D \in \mathbb{N}_0$, every polynomial $p \in \mathcal{G}$ of degree $D$ is in $\mathcal{F}_{K(D),D}^{\mathrm{TFN}}$.*
2. *Every continuous function $f \in \mathcal{G}$ can be approximated uniformly on compact sets by functions in $\bigcup_{D \in \mathbb{N}_0} \mathcal{F}_{K(D),D}^{\mathrm{TFN}}$.*

**Spherical networks.** Instead of intermediate $SO(3)$ representations we now switch to spherical representations, which are functions on the sphere $\boldsymbol{H} : S^2 \to \mathbb{R}$. We define the set of functions $\mathcal{F}_{K(D),D}^{\mathrm{sphere}}$ analogously to $\mathcal{F}_{K(D),D}^{\mathrm{TFN}}$. However, for $\mathcal{F}_{\mathrm{feat}}^{\mathrm{sphere}}(D)$ we use

$$\tilde{\boldsymbol{H}}_a^{\mathrm{sphere}}(\boldsymbol{X}, \boldsymbol{H})(\hat{\boldsymbol{r}}) = \theta \boldsymbol{H}_a(\hat{\boldsymbol{r}}) + \sum_{b \in \mathcal{N}_a} F_{\mathrm{sphere}}(\boldsymbol{x}_b - \boldsymbol{x}_a, \hat{\boldsymbol{r}}) \boldsymbol{H}_b(\hat{\boldsymbol{r}}), \tag{3}$$

with the filter function $F_{\mathrm{sphere}}(\boldsymbol{x}, \hat{\boldsymbol{r}}) = \sum_{l,m} R^{(l)}(x) \Re[Y_m^{(l)*}(\hat{\boldsymbol{x}}) Y_m^{(l)}(\hat{\boldsymbol{r}})]$, using the real part $\Re$ of the complex spherical harmonics $Y_m^{(l)}$. The set of pooling functions for invariant predictions is

$$\mathcal{F}_{\mathrm{pool}}^{\mathrm{sphere}} = \{f \mid f(\boldsymbol{H}) = \theta_{\mathrm{pool}} \int_{S^2} \boldsymbol{H}(\hat{\boldsymbol{r}}) \, \mathrm{d}\hat{\boldsymbol{r}}\}, \tag{4}$$

with the learnable parameter $\theta_{\mathrm{pool}}$. We obtain the universality theorem by showing the equivalence between this model and TFN for rotationally invariant functions. The proof is based on the connection between spherical harmonics and the Clebsch-Gordan coefficients [51, 3.7.72] (see App. A).

**Theorem 2.** *Consider the set of functions $\mathcal{G}'$ mapping $\mathbb{R}^{3 \times n + h \times n} \to W_{\mathrm{T}}^n$ that are equivariant to permutations and invariant to translations and rotations. For all $n \in \mathbb{N}$,*

1. *For $D \in \mathbb{N}_0$, every polynomial $p \in \mathcal{G}'$ of degree $D$ is in $\mathcal{F}_{K(D),D}^{\mathrm{sphere}}$.*
2. *Every continuous function $f \in \mathcal{G}'$ can be approximated uniformly on compact sets by functions in $\bigcup_{D \in \mathbb{N}_0} \mathcal{F}_{K(D),D}^{\mathrm{sphere}}$.*

Next, we extend Theorem 2 to rotationally *equivariant* functions. We do this by generalizing a recent result by Villar et al. [57] to obtain (see App. B):

**Theorem 3.** *Let $h \colon \mathbb{R}^{d \times n + h \times n} \to \mathbb{R}^{d \times n}$ be any function that is equivariant to permutations and rotations and invariant to translations. For all $a \in [1, n]$, let the set of relative vectors $\{\boldsymbol{x}_{ca} \mid c \in [1, n]\}$ not span a $(d-1)$-dimensional space. Then there are $n-1$ functions $f^{(c)} \colon \mathbb{R}^{d \times n + h \times n} \to \mathbb{R}^n$ such that*

$$\boldsymbol{h}_a(\boldsymbol{X}, \boldsymbol{H}) = \sum_{\substack{c=1 \\ c \neq a}}^n f_a^{(c)}(\boldsymbol{X}, \boldsymbol{H}) \boldsymbol{x}_{ca}, \tag{5}$$

*where $f^{(c)}$ is equivariant to permutations, but invariant to rotations and translations.*

This theorem lets us extend a rotationally invariant model to an equivariant one, while providing universality guarantees. Together, Theorem 2 and Theorem 3 (with $d = 3$) thus show that we can approximate any rotationally equivariant function using only representations on the $S^2$ sphere. We thus do not need $SO(3)$ representations, spin-weighted spherical harmonics [22], triplet embeddings, or complex-valued functions. This result puts theory back in line with practice, where the best results are currently achieved without relying on these more expensive representations [54].

## 4 From spherical representations to directional message passing

**Directional representations.** To use spherical representations in a model we first need to find a tractable description. Instead of using spherical harmonics, we propose to sample the representations

in specific directions $\hat{\boldsymbol{r}}_i$. If we look at recent models, we see that they implicitly use the directions to each atom's neighbors for this purpose, i.e. they embed the edges in the molecule's graph. These directions define an *equivariant* mesh that circumvents the aliasing effects that would arise from fixed grids [36]. Schütt et al. [54] flexibly define the directional mesh in each layer by aggregating directions, while Gasteiger et al. [29] and others use a fixed mesh for each atom. We can refine this mesh of directions e.g. by using more neighbors or by interpolating between directions. The approximation error of this directional mesh is related to the spherical harmonic expansion via the mesh norm and the separating distance between directions [33, 34]. Note that depending on the discretization scheme the resulting mesh might not provide a universal approximation guarantee.

Eq. (3) only defines the relationship for a fixed direction, while models commonly use different directional meshes for the input and output. To incorporate this we add a convolution with a learned filter $F_2$, which can only improve the model's expressiveness. Since the input and output are spherical functions, the used filter $F_2$ has to be *zonal*, i.e. it can only depend on one angle. This can be expressed as [17]

$$\tilde{\boldsymbol{H}}_a^{\mathrm{dir}}(\boldsymbol{X}, \boldsymbol{H})(\hat{\boldsymbol{r}}_o) = \theta \boldsymbol{H}_a(\hat{\boldsymbol{r}}_o) + \int_{\mathrm{SO}(3)} \sum_{b \in \mathcal{N}_a} F_{\mathrm{sphere}}(\boldsymbol{x}_{ba}, \boldsymbol{R}\hat{\boldsymbol{n}}) \sum_{i \in \mathcal{R}_b} \boldsymbol{H}_{bi} \delta(\boldsymbol{R}\hat{\boldsymbol{n}} - \hat{\boldsymbol{r}}_i) F_2(\boldsymbol{R}^{-1}\hat{\boldsymbol{r}}_o) \, \mathrm{d}\boldsymbol{R}$$

$$= \theta \boldsymbol{H}_a(\hat{\boldsymbol{r}}_o) + \sum_{b \in \mathcal{N}_a} \sum_{i \in \mathcal{R}_b} F_{\mathrm{sphere}}(\boldsymbol{x}_{ba}, \hat{\boldsymbol{r}}_i) \boldsymbol{H}_{bi} F_2(\measuredangle \hat{\boldsymbol{r}}_o \hat{\boldsymbol{r}}_i),$$

$$(6)$$

where $\mathcal{R}_b$ denotes the directional mesh of atom $b$ with mesh directions denoted by $\hat{\boldsymbol{r}}_i$, and $\hat{\boldsymbol{r}}_o$ specifies the output direction. The integral vanishes due to the Dirac delta $\delta$.

**General filters.** To see the relationship to GNNs we furthermore need to generalize the filter $F_{\mathrm{sphere}}(\boldsymbol{x}_{ba}, \hat{\boldsymbol{r}}_i)$. This filter only depends on the angle $\measuredangle \hat{\boldsymbol{r}}_i \hat{\boldsymbol{x}}_{ba}$ since it is rotationally invariant:

**Lemma 1.** $F_{\mathrm{sphere}}(\boldsymbol{R}\boldsymbol{x}, \boldsymbol{R}\hat{\boldsymbol{r}}) = F_{\mathrm{sphere}}(\boldsymbol{x}, \hat{\boldsymbol{r}})$ *for any rotation matrix* $\boldsymbol{R}$.

We can therefore substitute $F_{\mathrm{sphere}}$ with a general learnable filter $F_1$ that is parametrized by this relative angle. Since $F_{\mathrm{sphere}}$ arises as a special case we do not lose expressivity. We thus obtain

$$\tilde{\boldsymbol{H}}_a^{\mathrm{gem}}(\boldsymbol{X}, \boldsymbol{H})(\hat{\boldsymbol{r}}_o) = \theta \boldsymbol{H}_a(\hat{\boldsymbol{r}}_o) + \sum_{b \in \mathcal{N}_a} \sum_{i \in \mathcal{R}_b} F_1(x_{ba}, \measuredangle \hat{\boldsymbol{r}}_i \hat{\boldsymbol{x}}_{ba}) F_2(\measuredangle \hat{\boldsymbol{r}}_o \hat{\boldsymbol{r}}_i) \boldsymbol{H}_{bi}. \qquad (7)$$

We have now arrived at a message passing scheme that has universal approximation guarantees and is only based on relative directional information. To see the connection to GNNs we interpret these discretized spherical representations as edge embeddings pointing towards $\hat{\boldsymbol{r}}_o$ and $\hat{\boldsymbol{r}}_i$. Eq. (7) then corresponds to two-hop message passing between the edge embeddings of $\hat{\boldsymbol{r}}_o$ and $\hat{\boldsymbol{r}}_i$ via the edge $\hat{\boldsymbol{x}}_{ba}$. Interestingly, the central learnable part of Eq. (7) is the product of the filters $F_1(x_{ba}, \measuredangle \hat{\boldsymbol{r}}_i \hat{\boldsymbol{x}}_{ba})$ and $F_2(\measuredangle \hat{\boldsymbol{r}}_o \hat{\boldsymbol{r}}_i)$ with the input representation, which is strikingly similar to the Hadamard product used in modern GNNs [28, 53] – except that these only use one-hop message passing.

## 5  Geometric message passing

**Geometric representation.** We now develop a specific two-hop message passing scheme based on Eq. (7). We use embeddings based on interatomic directions, and embed all atom pairs with distance $x_{ca} \leq c_{\mathrm{emb}}$. $\hat{\boldsymbol{r}}_o$ and $\hat{\boldsymbol{r}}_i$ are thus instantiated as the interatomic directions $\hat{\boldsymbol{x}}_{ca}$ and $\hat{\boldsymbol{x}}_{db}$. We denote directional embeddings as $\boldsymbol{m}_{ca} = \boldsymbol{H}_a(\hat{\boldsymbol{x}}_{ca})$. Message passing is thus based on quadruplets of atoms – two atoms are interacting ($a$ and $b$) and two atoms define the directions ($c$ and $d$). We denote the angle between directions by $\varphi_{abd} = \measuredangle \hat{\boldsymbol{x}}_{ab} \hat{\boldsymbol{x}}_{db}$. To improve empirical performance we additionally use the dihedral angle $\theta_{cabd} = \measuredangle \hat{\boldsymbol{x}}_{ca} \hat{\boldsymbol{x}}_{db} \perp \hat{\boldsymbol{x}}_{ba}$ and substitute $\measuredangle \hat{\boldsymbol{r}}_o \hat{\boldsymbol{r}}_i = \measuredangle \hat{\boldsymbol{x}}_{ca} \hat{\boldsymbol{x}}_{db}$ with $\varphi_{cab}$. Fig. 1 illustrates the three angles $\varphi_{cab}$, $\varphi_{abd}$, and $\theta_{cabd}$ we use for updating the embedding $\boldsymbol{m}_{ca}$ based on $\boldsymbol{m}_{db}$. To ensure that all angles are well-defined we exclude overlapping atom quadruplets, i.e. $a \neq b \neq c \neq d$. We represent the relative directional information using spherical Fourier-Bessel

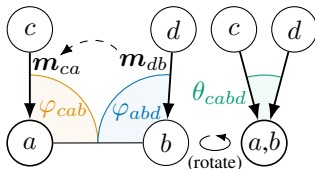

Figure 1: Angles used in geometric message passing. The dihedral angle $\theta_{cabd}$ becomes visible when rotating the molecule so that atoms $a$ and $b$ lie on top of each other (right).

bases with polynomial radial envelopes to ensure smoothly differentiable predictions, as proposed by Gasteiger et al. [29]. We split the basis into three parts to incorporate all available geometric information. Before the envelope, these are:

$$\tilde{e}_{\text{RBF},n}(x_{db}) = \sqrt{\frac{2}{c_{\text{emb}}}} \frac{\sin(\frac{n\pi}{c_{\text{emb}}} x_{db})}{x_{db}}, \tag{8}$$

$$\tilde{e}_{\text{CBF},ln}(x_{ba}, \varphi_{abd}) = \sqrt{\frac{2}{c_{\text{int}}^3 j_{l+1}^2(z_{ln})}} j_l(\frac{z_{ln}}{c_{\text{int}}} x_{ba}) Y_{l0}(\varphi_{abd}), \tag{9}$$

$$\tilde{e}_{\text{SBF},lmn}(x_{ca}, \varphi_{cab}, \theta_{cabd}) = \sqrt{\frac{2}{c_{\text{emb}}^3 j_{l+1}^2(z_{ln})}} j_l(\frac{z_{ln}}{c_{\text{emb}}} x_{ca}) Y_{lm}(\varphi_{cab}, \theta_{cabd}), \tag{10}$$

with the interaction cutoff $c_{\text{int}}$, the spherical Bessel functions $j_l$, and the $n$-th root of the $l$-order Bessel function $z_{ln}$. Note that Gasteiger et al. [29] only used the first two parts $e_{\text{RBF}}$ and $e_{\text{CBF}}$. These representations are then transformed using two linear layers to obtain the filter $F$. In order to maintain a smoothly differentiable cutoff we cannot use a bias in this transformation. Altogether, the core geometric message passing scheme is

$$\tilde{m}_{ca} = \sum_{\substack{b \in \mathcal{N}_a^{\text{int}} \backslash \{c\}, \\ d \in \mathcal{N}_b^{\text{emb}} \backslash \{a,c\}}} \Big( (\boldsymbol{W}_{\text{SBF1}} \boldsymbol{e}_{\text{SBF}}(x_{ca}, \varphi_{cab}, \theta_{cabd}))^T \mathbf{W}((\boldsymbol{W}_{\text{CBF2}} \boldsymbol{W}_{\text{CBF1}} \boldsymbol{e}_{\text{CBF}}(x_{ba}, \varphi_{abd})) \\ \odot (\boldsymbol{W}_{\text{RBF2}} \boldsymbol{W}_{\text{RBF1}} \boldsymbol{e}_{\text{RBF}}(x_{db})) \odot \boldsymbol{m}_{db}) \Big), \tag{11}$$

where $\boldsymbol{W}$ denotes a weight matrix, $\mathbf{W}$ denotes a weight tensor. The first weight matrix of each representation part has a small output dimension. This causes a bottleneck that improves generalization.

**Symmetric message passing.** Whenever we have a directional embedding $\boldsymbol{m}_{ca}$, we also have the opposing embedding $\boldsymbol{m}_{ac}$, since both are based on the same cutoff $c_{\text{emb}}$. Whether we associate the embedding $\boldsymbol{m}_{ca}$ or $\boldsymbol{m}_{ac}$ with atom $a$ is arbitrary. A more principled approach is to *jointly* interpret both embeddings as a representation of the atom pair $a$ and $c$. In this view, an update to $\boldsymbol{m}_{ca}$ should also influence $\boldsymbol{m}_{ac}$. This would normally require executing the above message passing scheme twice, once for updating $\boldsymbol{m}_{ca}$ based on $\boldsymbol{m}_{db}$, and once for updating $\boldsymbol{m}_{ac}$ based on $\boldsymbol{m}_{db}$. We propose to circumvent this double execution by calculating the update (Eq. (11)) only once and then using it for both $\boldsymbol{m}_{ca}$ and $\boldsymbol{m}_{ac}$. To preserve the distinction between the two directions and ensure that $\boldsymbol{m}_{ca} \neq \boldsymbol{m}_{ac}$, we transform the two updates using two separate learnable weight matrices. One single message passing update thus carries information for both embeddings, which is then dissected by the two weight matrices. In practice, this only requires a simple re-indexing operation that maps the edge $ca$ to $ac$.

**Efficient bilinear layer.** The whole message passing scheme, i.e. basis transformation, neighbor aggregation, and bilinear layer, only use linear functions. We can therefore freely optimize the order of summation without changing the result, as proposed by Wu et al. [59] (see App. D for details). Doing so can provide a faster and more memory-efficient model, reducing memory usage by 50 % even for Hadamard products. Moreover, since everything is based on efficient matrix products, this allows us to use the bilinear layer at practically no additional cost compared to a Hadamard product. Note that this requires using padded matrices instead of the usual gather-scatter operations to prevent excessively large intermediate results.

## 6   GemNet: Geometric message passing neural network

**GemNet.** The geometric message passing neural network (GemNet) is a significantly refined architecture based on DimeNet$^{++}$ [28, Hippocratic license 2.1]. GemNet predicts the molecular energy $E$ and forces $\boldsymbol{F} \in \mathbb{R}^{3 \times n}$ based on the atomic positions $\boldsymbol{X} \in \mathbb{R}^{3 \times n}$ and the atomic numbers $\boldsymbol{z} \in \mathbb{N}^n$. The architecture is illustrated in Fig. 2. A comprehensive version with low-level layers and hyperparameters is described in App. F. GemNet was developed on the COLL dataset, but generalizes to other datasets such as MD17 without architectural changes. Every change we propose either improves model performance or reduces model complexity. For example, GemNet uses no biases since we found them to be irrelevant or even detrimental to accuracy. We show the impact of the most relevant changes via ablation studies in Sec. 7.

**Interactions.** GemNet incorporates three forms of interactions. The first is geometric message passing, as described in Sec. 5. The second is a one-hop form of geometric message passing. This

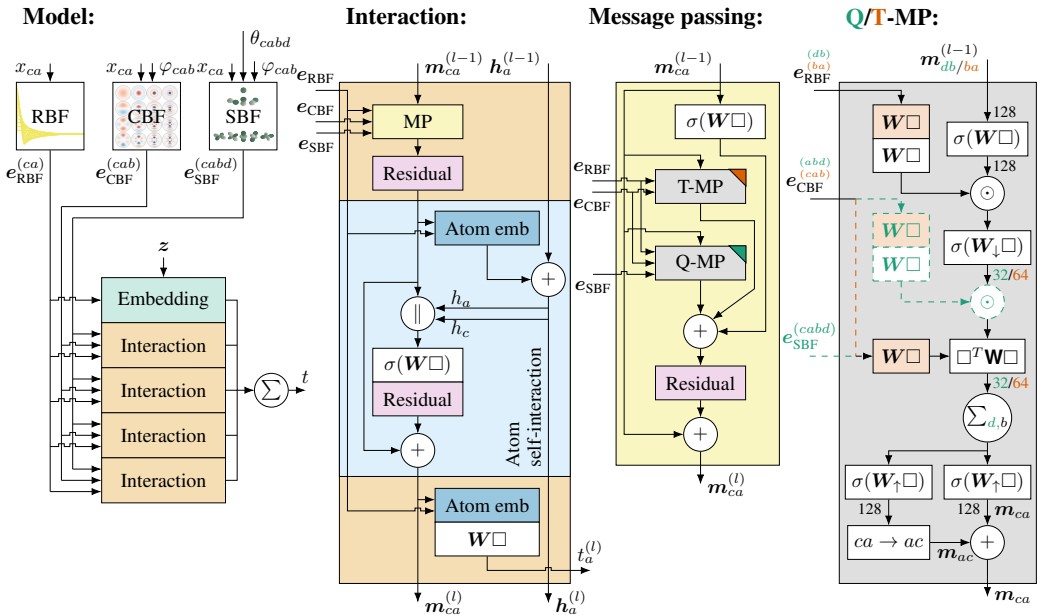

Figure 2: The GemNet architecture (comprehensive version in App. F). $\square$ denotes the layer's input, $\|$ concatenation, and $\sigma$ a non-linearity. Directional embeddings $\boldsymbol{m}_{ca}$ are updated using three forms of interaction: Two-hop geometric message passing (Q-MP), one-hop geometric message passing (T-MP), and atom self-interactions. Differences between Q-MP and T-MP are denoted by colors and dashed lines.

interaction uses a single cutoff $c = c_{\text{emb}}$ and passes messages only between directional embeddings pointing towards the same atom, similarly to DimeNet [29]. This provides both angle-based pair interactions and atom self-interactions, thanks to the symmetric message passing scheme described in Sec. 5. The third interaction is a pure atom self-interaction based on atom embeddings. We first aggregate the directional embeddings of one atom to obtain an atom embedding. We then use this atom embedding to update all directional embeddings. We found all three interaction forms to be beneficial, and show this in our ablation studies.

**Stabilizing activation variance.** The variance of activations in a model is usually stabilized using normalization methods, which has various positive effects on training [16, 41, 52]. However, they also have multiple undesirable side effects, especially in the context of molecular regression. Batch normalization introduces correlations between separate molecules and atoms. Layer normalization forces all activation scales to be constant, while atomic interactions actually cover a large range of scales – directly bonding atoms have a substantially stronger interaction than atoms at a long range. To circumvent these issues, we stabilize GemNet's variance by introducing constant scaling factors, as suggested by Brock et al. [6]. We found that the activation variance is primarily impacted by four components: Skip connections, non-linearities, message aggregation, and Hadamard/bilinear layers. The two summands in a skip connection $y = x + f(x)$ have no covariance at initialization due to random weight matrices. We can thus remove its impact by scaling the output by $1/\sqrt{2}$. We remove the non-linearity's impact by scaling its output with a gain of $\gamma = 1/0.6$ for SiLU, similarly to [35]. Note that we do not center SiLU's output but instead choose a slightly lower $\gamma$ to account for mean shift. Additionally, we standardize the weight matrices to have exactly zero mean and $1/\text{fan-in}$ variance. The sum aggregation and Hadamard/bilinear layers have a more complex impact on the variance, which we cannot determine a priori (see App. E for details). We therefore estimate the variance after these layers based on random batches of data. We then rescale their output accordingly to obtain roughly the variance of the layer input at initialization. These simple empirical scaling factors are sufficient to keep the activation variance roughly constant (see Fig. 3). We found that other measures such as adaptive gradient clipping [7], scaled weight standardization [6], or weighting the residual block with zero at initialization [16] are not beneficial for model accuracy.

Table 1: MAE on COLL, in meV/Å and meV. GemNet is 34 % more accurate for forces. The higher energy error is due to its lower loss weight.

|  | Forces | Energy |
|---|---|---|
| SchNet | 172 | 198 |
| DimeNet$^{++}$ | 40 | **47** |
| GemNet-Q | **26.4** | 53 |
| GemNet-T | 31.6 | 60 |
| GemNet-dQ | 38.1 | 60 |
| GemNet-dT | 43.1 | 55 |

Table 2: Force MAE for MD17@CCSD in meV/Å. GemNet outperforms previous methods by 44 % on average.

|  | sGDML | NequIP | GemNet-Q | GemNet-T |
|---|---|---|---|---|
| Aspirin | 33.0 | 14.7 | 10.4 | **10.3** |
| Benzene | 1.7 | 0.8 | **0.7** | **0.7** |
| Ethanol | 15.2 | 9.4 | **3.1** | **3.1** |
| Malonaldehyde | 16.0 | 16.0 | 6.0 | **5.9** |
| Toluene | 9.1 | 4.4 | **2.5** | 2.7 |

**GemNet-Q and GemNet-T.** Geometric message passing is comparatively expensive since it is based on quadruplets of atoms. Its runtime thus scales with $\mathcal{O}(nk_{\text{int}}k_{\text{emb}}^2)$, where $k_{\text{int}}$ is the number of interacting neighbors, and $k_{\text{emb}}$ is the number of embedded directions. For this reason we investigate two message passing models in our experiments – one with two-hop geometric message passing (GemNet-Q) and one using only the two cheaper forms of interaction (GemNet-T). Their complexities are $\mathcal{O}(nk_{\text{int}}k_{\text{emb}}^2)$ and $\mathcal{O}(nk_{\text{emb}}^2)$, respectively. Note that GemNet-T is thus a direct ablation of the two-hop message passing scheme implied by our theoretical results.

**Direct force predictions.** GemNet predicts forces by calculating $\boldsymbol{F}_a = -\partial E/\partial \boldsymbol{x}_a$ via backpropagation. This form of calculation guarantees a conservative force field, which is important for the stability of simulations. However, by using Eq. (5) we can also directly predict forces and other vector quantities. This essentially means predicting a magnitude for each directional embedding and then summing up over the vectors defined by this magnitude and the embedding's associated direction, similarly to Park et al. [47]. We denote this variant as *GemNet-dQ* and *GemNet-dT*. Interestingly, GemNet is thus able to generate rotationally *equivariant* predictions despite only using *invariant* representations. Direct predictions substantially accelerate the model, especially for training. For most datasets, the resulting accuracy is on par with most previous models, but significantly worse than GemNet's accuracy via backpropagation. However, this is not true for OC20, where we found GemNet-dT to converge faster and perform on par with GemNet-T.

**Limitations.** GemNet is focused on one specific, important task: Predictions for molecular simulations. We do not make any statements regarding its performance beyond this scope. The GemNet architecture might seem more complex than some previous models, due to its larger variety of interactions and blocks. However, its number of parameters and training or inference time is actually on par with previous models. Two-hop message passing introduces significant computational overhead. We mitigate this effect with a down-projection layer and additionally introduce the ablated GemNet-T model. This model performs surprisingly well on MD17, but not on COLL. This suggests that one-hop message passing is expressive enough for some practical use cases, but two-hop message passing gives an advantage for the more challenging task of fitting multiple molecules at once.

**Societal impacts.** Accelerating molecular simulations can have positive effects in a wide range of applications in physics and chemistry. At the same time, however, this can be used for malicious purposes such as developing chemical agents or weapons. To the best of our knowledge, this work does not promote these use cases more than regular chemistry research does. To somewhat mitigate negative effects we will publish our source code under the Hippocratic license [19].

## 7 Experiments

**Experimental setup.** We evaluate our model on four molecular dynamics datasets. COLL [28, CC-BY 4.0] consists of configurations taken from molecular collisions of different small organic molecules. MD17 [9] consists of configurations of multiple separate, thermalized molecules, considering only one molecule at a time. MD17@CCSD [10] uses the same setup, but calculates the forces using the more accurate and expensive CCSD or CCSD(T) method. The open catalyst (OC20) dataset [8, CC-BY 4.0] consists of energy relaxation trajectories of solid catalysts with adsorbate molecules. This dataset is split into three tasks: (1) Structure to energy and forces (S2EF), which is the same task as used by the COLL and MD17 datasets, (2) initial structure to relaxed structure (IS2RS), where

Table 3: Force MAE for MD17 in meV/Å. GemNet outperforms all previous methods by a wide margin, on average by 41 %.

| | Kernel methods | | GNNs | | | | **GemNet** | |
| | sGDML | FCHL19 | DimeNet | SphereNet | NequIP | PaiNN | GemNet-Q | GemNet-T |
|---|---|---|---|---|---|---|---|---|
| Aspirin | 29.5 | 20.7 | 21.6 | 18.6 | 15.1 | 14.7 | **9.4** | 9.5 |
| Benzene[9] | - | - | 8.1 | 7.7 | 8.1 | - | **6.3** | **6.3** |
| Benzene[10] | 2.6 | - | - | - | 2.3 | - | 1.5 | **1.4** |
| Ethanol | 14.3 | 5.9 | 10.0 | 9.0 | 9.0 | 9.7 | 3.8 | **3.7** |
| Malonaldehyde | 17.8 | 10.6 | 16.6 | 14.7 | 14.6 | 14.9 | 6.9 | **6.7** |
| Naphthalene | 4.8 | 6.5 | 9.3 | 7.7 | 4.2 | 3.3 | **2.2** | 2.4 |
| Salicylic acid | 12.1 | 9.6 | 16.2 | 15.6 | 10.3 | 8.5 | **5.4** | 5.5 |
| Toluene | 6.1 | 8.8 | 9.4 | 6.7 | 4.4 | 4.1 | **2.6** | **2.6** |
| Uracil | 10.4 | 4.6 | 13.1 | 11.6 | 7.5 | 6.0 | 4.5 | **4.2** |

Table 4: Results for the three tasks of the open catalyst dataset (OC20), averaged across its four test sets. GemNet outperforms all previous methods in all measures, on average by 20 %. *DimeNet$^{++}$-large uses separate models for energy and force prediction for IS2RE.

| | S2EF | | | IS2RS | | IS2RE |
| | Energy MAE | Force MAE | Force cos | AFbT | ADwT | Energy MAE |
| | meV ↓ | meV/Å ↓ | ↑ | % ↑ | % ↑ | meV ↓ |
|---|---|---|---|---|---|---|
| ForceNet-large | - | 31.2 | 0.520 | 12.7 | 49.6 | - |
| DimeNet$^{++}$-large* | - | 31.3 | 0.544 | 21.8 | 51.7 | 559.1 |
| SpinConv | 336.3 | 29.7 | 0.539 | 16.7 | 53.6 | 434.3 |
| **GemNet-dT** | **292.4** | **24.2** | **0.616** | **27.6** | **58.7** | **399.7** |

an energy optimization is carried out based on the model's predictions and we measure how close the final structure is to the true relaxed structure (average distance within threshold, ADwT) and whether the final forces are close to zero (average forces below threshold, AFbT), and (3) initial structure to relaxed energy (IS2RE), where we predict the energy of the relaxed structure, based on an energy optimization starting at the initial structure. All presented OC20 models are trained on the S2EF data. Following the setup of Batzner et al. [4], we use 1000 training and validation configurations for MD17, and 950 training and 50 validation configurations for MD17@CCSD. We focus on force predictions and use a high force loss weight since they determine the accuracy of molecular simulations. We measure the mean absolute error (MAE), averaged over all samples, atoms, and components. We compare with the results reported by several state-of-the-art models: sGDML [10], FCHL19 [12], SchNet [53], DimeNet [29], DimeNet$^{++}$ [28], SphereNet [39], NequIP [4], PaiNN [54], ForceNet [32], and SpinConv [55]. For further details see App. G.

**Results.** Tables 1 to 4 show that GemNet-T and GemNet-Q consistently perform best on all molecular dynamics datasets investigated – and by a large margin. This is true both in comparison to previous GNNs and for kernel methods – despite the latter typically being more sample efficient. The improvements are largest for chain-like molecules, such as ethanol and malonaldehyde. These molecules are the most challenging since they exhibit a wide range of movement. GemNet even performs better than some previous models that were trained with 50x more training samples. For example, it performs better than SchNet with 50 000 training samples on six out of eight MD17 molecules (see Table 9). Interestingly, the two-hop message passing scheme implied by our theoretical results (GemNet-Q) yields significant improvements on COLL, but performs approximately on par with the ablated GemNet-T on MD17. To investigate this disagreement we trained GemNet on a combined dataset of all MD17 molecules. Table 13 shows that GemNet-Q again performs better than GemNet-T in this setting. These results suggest that regular MD17 is too simple to show the benefits of two-hop message passing. It seems to be particularly important in more difficult settings that cover a large variety of configurations and molecules.

**Computational aspects.** GemNet-Q is roughly two times slower than GemNet-T (see Table 14). Thanks to the efficient aggregation, GemNet with bilinear layers is as fast as with regular Hadamard products. Efficient aggregation also reduces the memory usage for regular Hadamard products by around 50 % (from 4.1GB to 2.2GB for a batch of 32 Toluene molecules). Note that GemNet has not

been optimized for runtime and can likely be accelerated substantially. GemNet-Q uses 2.2M and GemNet-T 1.9M parameters, which is comparable to previous models such as DimeNet$^{++}$, which uses 1.9M parameters. See App. I for further details.

**Direct force prediction.** Directly predicting the forces accelerates training by four times on average and inference by 1.6 times on average in our experiments (see Table 14), while reducing memory consumption by roughly a factor of two. While using direct predictions instead of backpropagation increases the MAE by 44 % on COLL and by 48 % on MD17 (see Tables 1 and 7), they actually perform better on the S2EF task on OC20. This is likely due to OC20 being orders of magnitude larger than COLL and MD17. Whether to use direct predictions thus depends on the dataset and the application's computational requirements.

**Ablation studies.** We investigate the proposed architectural improvements on COLL in Table 5. The proposed symmetric message passing scheme yields significant accuracy improvements, as does using a bilinear layers instead of a Hadamard product. We also see that removing any of the three interaction forms described in Sec. 6 increases the error, showing that this combination is indeed beneficial. The proposed scaling factors also yield decent improvements, while regular layer normalization actually increases the error. Two-hop message passing yields the largest single improvement. Table 10 shows that our architectural improvements yield similar benefits for DimeNet$^{++}$. Overall, the error improvements are quite evenly distributed. This suggests that GemNet's improved performance is not due to one single change, but rather due to the full range of improvements proposed in this work.

Table 5: Ablation studies on COLL. Force MAE in meV/Å after 500 000 training steps. All proposed components yield significant improvements.

| Model | Forces |
|---|---|
| without symmetric message passing | 28.5 |
| Hadamard product instead of bilinear layer | 29.3 |
| without atom embedding updates | 28.3 |
| without one-hop message passing | 31.3 |
| without two-hop message passing | 32.4 |
| without scaling factors | 29.1 |
| use layer norm instead (without centering) | 33.3 |
| with bias | 27.2 |
| GemNet-Q | 27.0 |

## 8   Conclusion

In this work we proved the universality for GNNs using directional embeddings. We proposed geometric message passing based on these insights, and improved this scheme with symmetric message passing and efficient bilinear layers. We incorporated these improvements in the GemNet architecture, which substantially improves the error on various molecular dynamics datasets. We showed that all of the proposed enhancements yield significant performance improvements. Most of our proposed improvements are of independent interest for other molecular GNNs.

## Acknowledgments and Disclosure of Funding

We would like to thank Soledad Villar for help with proving Lemma B, as well as Nicholas Gao and Aleksandar Bojchevski for their invaluable feedback and comments.

This research was supported by the Deutsche Forschungsgemeinschaft (DFG) through the Emmy Noether grant GU 1409/2-1 and the TUM International Graduate School of Science and Engineering (IGSSE), GSC 81.

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
