# A Proof of Theorem 2

We prove the universal approximation theorem by showing the equivalence of TFN and our model. Complex spherical harmonics are related to Clebsch-Gordan coefficients via [51, 3.7.72]

$$Y_{m_i}^{(l_i)}(\hat{\boldsymbol{r}})Y_{m_f}^{(l_f)}(\hat{\boldsymbol{r}}) = \sum_{l_o,m_o} \sqrt{\frac{(2l_i+1)(2l_f+1)}{4\pi(2l_o+1)}} C_{(l_f,0),(l_i,0)}^{(l_o,0)} C_{(l_f,m_f),(l_i,m_i)}^{(l_o,m_o)} Y_{m_o}^{(l_o)}(\hat{\boldsymbol{r}}). \tag{12}$$

We now use the fact that multiplying a learnable function with a unitary matrix or a scalar does not change the resulting function space. We can therefore adapt Eq. (2) by substituting

$$C_{(l_f,m_f),(l_i,m_i)}^{(l_o,m_o)} \mapsto C(l_f,m_f,l_i,m_i,l_o,m_o) = \sqrt{\frac{(2l_i+1)(2l_f+1)}{4\pi(2l_o+1)}} C_{(l_f,0),(l_i,0)}^{(l_o,0)} C_{(l_f,m_f),(l_i,m_i)}^{(l_o,m_o)} \tag{13}$$

without impacting model expressivity. Since real spherical harmonics and complex (conjugate) spherical harmonics cover the same function space, we can furthermore substitute the filter with $F_m'^{(l)}(\boldsymbol{x}) = R^{(l)}(x)Y_m^{(l)*}(\hat{\boldsymbol{x}})$. Using the spherical harmonics expansion we therefore obtain

$$\tilde{\boldsymbol{H}}_a'(\boldsymbol{X},\boldsymbol{H}')(\hat{\boldsymbol{r}}) = \sum_{l_o,m_o} \tilde{\boldsymbol{H}}_{am_o}'^{(l_o)}(\boldsymbol{X},\boldsymbol{H}')Y_{m_o}^{(l_o)}(\hat{\boldsymbol{r}})$$

$$= \sum_{l_o,m_o} \left( \theta\boldsymbol{H}_{am_o}'^{(l_o)} + \sum_{l_f,m_f}\sum_{l_i,m_i} C(l_f,m_f,l_i,m_i,l_o,m_o) \sum_{b\in\mathcal{N}_a} F_{m_f}'^{(l_f)}(\boldsymbol{x}_{ba})\boldsymbol{H}_{bm_i}'^{(l_i)} \right) Y_{m_o}^{(l_o)}(\hat{\boldsymbol{r}})$$

$$= \theta\boldsymbol{H}_a'(\hat{\boldsymbol{r}}) + \sum_{l_f,m_f}\sum_{l_i,m_i}\sum_{b\in\mathcal{N}_a} F_{m_f}'^{(l_f)}(\boldsymbol{x}_{ba})Y_{m_f}^{(l_f)}(\hat{\boldsymbol{r}})\boldsymbol{H}_{bm_i}'^{(l_i)}Y_{m_i}^{(l_i)}(\hat{\boldsymbol{r}})$$

$$= \theta\boldsymbol{H}_a'(\hat{\boldsymbol{r}}) + \sum_{b\in\mathcal{N}_a} \left( \sum_{l_f,m_f} F_{m_f}'^{(l_f)}(\boldsymbol{x}_{ba})Y_{m_f}^{(l_f)}(\hat{\boldsymbol{r}}) \right) \left( \sum_{l_i,m_i} \boldsymbol{H}_{bm_i}'^{(l_i)}Y_{m_i}^{(l_i)}(\hat{\boldsymbol{r}}) \right)$$

$$= \theta\boldsymbol{H}_a'(\hat{\boldsymbol{r}}) + \sum_{b\in\mathcal{N}_a} F'(\boldsymbol{x}_{ba},\hat{\boldsymbol{r}})\boldsymbol{H}_b'(\hat{\boldsymbol{r}}). \tag{14}$$

These functions rely on complex-valued representations, while the output and $SO(3)$ representations are real-valued. However, we can restrict the representations to real values without changing the resulting function space. To see this, we look at the result's real component

$$\Re[\tilde{\boldsymbol{H}}_a'(\boldsymbol{X},\boldsymbol{H}')(\hat{\boldsymbol{r}})] = \theta\Re[\boldsymbol{H}_a'(\hat{\boldsymbol{r}})] + \sum_{b\in\mathcal{N}_a} \Re[F'(\boldsymbol{x}_{ba},\hat{\boldsymbol{r}})\boldsymbol{H}_b'(\hat{\boldsymbol{r}})]$$

$$= \theta\Re[\boldsymbol{H}_a'(\hat{\boldsymbol{r}})] + \sum_{b\in\mathcal{N}_a} (\Re[F'(\boldsymbol{x}_{ba},\hat{\boldsymbol{r}})]\Re[\boldsymbol{H}_b'(\hat{\boldsymbol{r}})] - \Im[F'(\boldsymbol{x}_{ba},\hat{\boldsymbol{r}})]\Im[\boldsymbol{H}_b'(\hat{\boldsymbol{r}})]). \tag{15}$$

The function space covered by $\Re[F'(\boldsymbol{x},\hat{\boldsymbol{r}})]$, and thus $\Re[\boldsymbol{H}'(\hat{\boldsymbol{r}})]$, is the same as $\Im[F'(\boldsymbol{x},\hat{\boldsymbol{r}})]$, and thus $\Im[\boldsymbol{H}'(\hat{\boldsymbol{r}})]$. We can therefore simply remove the imaginary part without changing the resulting function space, obtaining

$$\tilde{\boldsymbol{H}}_a^{\text{sphere}}(\boldsymbol{X},\boldsymbol{H})(\hat{\boldsymbol{r}}) = \theta\boldsymbol{H}_a(\hat{\boldsymbol{r}}) + \sum_{b\in\mathcal{N}_a} \Re[F'(\boldsymbol{x}_{ba},\hat{\boldsymbol{r}})]\boldsymbol{H}_b(\hat{\boldsymbol{r}})$$

$$= \theta\boldsymbol{H}_a(\hat{\boldsymbol{r}}) + \sum_{b\in\mathcal{N}_a} F_{\text{sphere}}(\boldsymbol{x}_{ba},\hat{\boldsymbol{r}})\boldsymbol{H}_b(\hat{\boldsymbol{r}}). \tag{16}$$

$\mathcal{F}_{\text{feat}}^{\text{sphere}}(D)$ thus spans the exact same space of embedding functions as $\mathcal{F}_{\text{feat}}^{\text{TFN}}(D)$, despite only using real functions on the $S^2$ sphere. However, we cannot span the full space of rotationally equivariant linear pooling functions, since equivariant linear functions on the $S^2$ sphere are limited to convolutions with zonal filters [21]. Fortunately, scalar pooling functions are limited to linear functions of the constant $l=0$ part. This is equivalent to integrating over the real-space spherical representation, as done in $\mathcal{F}_{\text{pool}}^{\text{sphere}}$. □

# B  Proof of Theorem 3

To prove this theorem we first introduce a proposition by Villar et al. [57].

**Proposition A** (Villar et al. [57])**.** *If $h$ is an $\mathrm{SO}(d)$-equivariant function $\mathbb{R}^{d \times n} \to \mathbb{R}^d$ of $n$ vector inputs $\boldsymbol{x}_1, \boldsymbol{x}_2, \ldots, \boldsymbol{x}_n$, then there are $n$ $\mathrm{SO}(d)$-invariant functions $f_c \colon \mathbb{R}^{d \times n} \to \mathbb{R}$ such that*

$$h(\boldsymbol{x}_1, \boldsymbol{x}_2, \ldots, \boldsymbol{x}_n) = \sum_{c=1}^{n} f^{(c)}(\boldsymbol{x}_1, \boldsymbol{x}_2, \ldots, \boldsymbol{x}_n) \boldsymbol{x}_c, \tag{17}$$

*except when $\boldsymbol{x}_1, \boldsymbol{x}_2, \ldots, \boldsymbol{x}_n$ span a $(d-1)$-dimensional space. In that case, there exist $\mathrm{O}(d)$-invariant functions $f_c \colon \mathbb{R}^{d \times n} \to \mathbb{R}$ such that*

$$h(\boldsymbol{x}_1, \boldsymbol{x}_2, \ldots, \boldsymbol{x}_n) = \sum_{c=1}^{n} f^{(c)}(\boldsymbol{x}_1, \boldsymbol{x}_2, \ldots, \boldsymbol{x}_n) \boldsymbol{x}_c + \sum_{S \in \binom{[n]}{d-1}} f^{(S)}(\boldsymbol{x}_1, \boldsymbol{x}_2, \ldots, \boldsymbol{x}_n) \boldsymbol{x}_S, \tag{18}$$

*where $[n] := \{1, \ldots, n\}$, $\binom{[n]}{d-1}$ is the set of all $(d-1)$-subsets of $[n]$, and $\boldsymbol{x}_S$ is the generalized cross product of vectors $\boldsymbol{x}_i$ with $i \in S$ (taken in ascending order).*

To extend Prop. A to our case, we need to restrict the functions to being translation-invariant and permutation-equivariant. We will only concern ourselves with the case where the vectors do not span a $(d-1)$-dimensional space. We start by considering translation-invariant functions, following the proof idea of Villar et al. [57, Lemma 7].

**Lemma A.** *Let $h$ be a translation-invariant and $\mathrm{SO}(d)$-equivariant function $\mathbb{R}^{d \times n} \to \mathbb{R}^d$ of $n$ vector inputs $\boldsymbol{x}_1, \boldsymbol{x}_2, \ldots, \boldsymbol{x}_n$. Let $\boldsymbol{x}_2 - \boldsymbol{x}_1, \ldots, \boldsymbol{x}_n - \boldsymbol{x}_1$ not span a $(d-1)$-dimensional space. Then there are $n-1$ translation- and $\mathrm{SO}(d)$-invariant functions $f_c \colon \mathbb{R}^{d \times n} \to \mathbb{R}$ such that*

$$h(\boldsymbol{x}_1, \boldsymbol{x}_2, \ldots, \boldsymbol{x}_n) = \sum_{c=2}^{n} f^{(c)}(\boldsymbol{x}_1, \boldsymbol{x}_2, \ldots, \boldsymbol{x}_n)(\boldsymbol{x}_c - \boldsymbol{x}_1). \tag{19}$$

*Proof.* Consider the $\mathrm{SO}(d)$-equivariant function $\tilde{h} \colon \mathbb{R}^{d \times (n-1)} \to \mathbb{R}^d$ with

$$h(\boldsymbol{x}_1, \boldsymbol{x}_2, \ldots, \boldsymbol{x}_n) = h(0, \boldsymbol{x}_2 - \boldsymbol{x}_1, \ldots, \boldsymbol{x}_n - \boldsymbol{x}_1) = \tilde{h}(\boldsymbol{x}_2 - \boldsymbol{x}_1, \ldots, \boldsymbol{x}_n - \boldsymbol{x}_1). \tag{20}$$

Due to Prop. A we have

$$\tilde{h}(\boldsymbol{x}_2 - \boldsymbol{x}_1, \ldots, \boldsymbol{x}_n - \boldsymbol{x}_1) = \sum_{c=2}^{n} \tilde{f}^{(c)}(\boldsymbol{x}_2 - \boldsymbol{x}_1, \ldots, \boldsymbol{x}_n - \boldsymbol{x}_1)(\boldsymbol{x}_c - \boldsymbol{x}_1), \tag{21}$$

with the $\mathrm{SO}(d)$-equivariant function $\tilde{f}^{(c)}$. If we now substitute $\tilde{f}^{(c)}$ with the $\mathrm{SO}(d)$-equivariant and translation-invariant function $f^{(c)}$, i.e.

$$\tilde{f}^{(c)}(\boldsymbol{x}_2 - \boldsymbol{x}_1, \ldots, \boldsymbol{x}_n - \boldsymbol{x}_1) = f^{(c)}(0, \boldsymbol{x}_2 - \boldsymbol{x}_1, \ldots, \boldsymbol{x}_n - \boldsymbol{x}_1) = f^{(c)}(\boldsymbol{x}_1, \boldsymbol{x}_2, \ldots, \boldsymbol{x}_n), \tag{22}$$

we obtain

$$h(\boldsymbol{x}_1, \boldsymbol{x}_2, \ldots, \boldsymbol{x}_n) = \sum_{c=2}^{n} f^{(c)}(\boldsymbol{x}_1, \boldsymbol{x}_2, \ldots, \boldsymbol{x}_n)(\boldsymbol{x}_c - \boldsymbol{x}_1). \tag{23}$$

$\square$

Next, we extend this result to permutation-equivariant functions.

**Lemma B.** *Let $h$ be a translation-invariant, and permutation and $\mathrm{SO}(d)$-equivariant function $\mathbb{R}^{d \times n} \to \mathbb{R}^{d \times n}$ of $n$ vector inputs $\boldsymbol{x}_1, \boldsymbol{x}_2, \ldots, \boldsymbol{x}_n$. Let $\boldsymbol{x}_2 - \boldsymbol{x}_1, \ldots, \boldsymbol{x}_n - \boldsymbol{x}_1$ not span a $(d-1)$-dimensional space. Then there are $n-1$ translation- and $\mathrm{SO}(d)$-invariant, and permutation-equivariant functions $f_c \colon \mathbb{R}^{d \times n} \to \mathbb{R}^n$ such that*

$$h(\boldsymbol{x}_1, \boldsymbol{x}_2, \ldots, \boldsymbol{x}_n) = \sum_{c=2}^{n} f^{(c)}(\boldsymbol{x}_1, \boldsymbol{x}_2, \ldots, \boldsymbol{x}_n)(\boldsymbol{x}_c - \boldsymbol{x}_1). \tag{24}$$

*Proof.* Permutation equivariance implies that for all $s$ and $t$ (w.l.o.g. $s < t$)

$$h_s(\ldots, \boldsymbol{x}_s, \ldots, \boldsymbol{x}_t, \ldots) = h_t(\ldots, \boldsymbol{x}_t, \ldots, \boldsymbol{x}_s, \ldots). \tag{25}$$

Due to Lemma A we have

$$h_s(\ldots, \boldsymbol{x}_s, \ldots, \boldsymbol{x}_t, \ldots) = \sum_{c=2}^{n} f_s^{(c)}(\ldots, \boldsymbol{x}_s, \ldots, \boldsymbol{x}_t, \ldots)(\boldsymbol{x}_c - \boldsymbol{x}_1), \tag{26}$$

$$= h_t(\ldots, \boldsymbol{x}_t, \ldots, \boldsymbol{x}_s, \ldots) = \sum_{c=2}^{n} f_t^{(c)}(\ldots, \boldsymbol{x}_t, \ldots, \boldsymbol{x}_s, \ldots)(\boldsymbol{x}_c - \boldsymbol{x}_1), \tag{27}$$

with $n - 1$ SO($d$)- and translation-invariant functions $f^{(c)} \colon \mathbb{R}^{d \times n} \to \mathbb{R}^n$. We can solve this equation by choosing

$$f_s^{(c)}(\ldots, \boldsymbol{x}_s, \ldots, \boldsymbol{x}_t, \ldots) = f_t^{(c)}(\ldots, \boldsymbol{x}_t, \ldots, \boldsymbol{x}_s, \ldots), \tag{28}$$

i.e. permutation-equivariant functions $f^{(c)}$. $\qquad\square$

Finally, to bring Lemma B to the form presented in the theorem, we first observe that adding scalar inputs $\boldsymbol{H}$ does not affect the proofs in this section. Second, we observe that subtracting by $\boldsymbol{x}_1$ in Eq. (20) is arbitrary. To bring this more in line with GNNs we can instead subtract the input of each $h_a$ by $\boldsymbol{x}_a$. This yields

$$h_a(\boldsymbol{X}, \boldsymbol{H}) = \sum_{\substack{c=1 \\ c \neq a}}^{n} f_a^{(c)}(\boldsymbol{X}, \boldsymbol{H})(\boldsymbol{x}_c - \boldsymbol{x}_a). \tag{29}$$

$\qquad\square$

## C   Proof of Lemma 1

Using the fact that the Wigner-D matrix is unitary, we obtain for any rotation matrix $\boldsymbol{R}$:

$$\begin{aligned}
F_{\text{sphere}}(\boldsymbol{R}\boldsymbol{x}, \boldsymbol{R}\hat{\boldsymbol{r}}) &= \sum_{l,m} R^{(l)}(x) \Re[Y_m^{(l)*}(\boldsymbol{R}\hat{\boldsymbol{x}}) Y_m^{(l)}(\boldsymbol{R}\hat{\boldsymbol{r}})] \\
&= \sum_{l,m,m',m''} R^{(l)}(x) \Re[Y_{m'}^{(l)*}(\hat{\boldsymbol{x}}) D_{m,m'}^{(l)*}(\boldsymbol{R}) D_{m,m''}^{(l)}(\boldsymbol{R}) Y_{m''}^{(l)}(\hat{\boldsymbol{r}})] \\
&= \sum_{l,m',m''} R^{(l)}(x) \Re[Y_{m'}^{(l)*}(\hat{\boldsymbol{x}}) \delta_{m',m''} Y_{m''}^{(l)}(\hat{\boldsymbol{r}})] \\
&= \sum_{l,m'} R^{(l)}(x) \Re[Y_{m'}^{(l)*}(\hat{\boldsymbol{x}}) Y_{m'}^{(l)}(\hat{\boldsymbol{r}})] = F_{\text{sphere}}(\boldsymbol{x}, \hat{\boldsymbol{r}}).
\end{aligned} \tag{30}$$

$\qquad\square$

## D   Efficient message passing

For clarity we demonstrate how to optimize the summation order using the simpler one-hop message passing. For a regular Hadamard product we reorder the sums as

$$\begin{aligned}
\boldsymbol{m}_{(ca)i} &= \sum_{b \in \mathcal{N}_a \backslash \{c\}} \left( \boldsymbol{W}^{(2)} \boldsymbol{W}^{(1)} \boldsymbol{e}_{\text{CBF}}(x_{ca}, \varphi_{bac}) \right)_i \boldsymbol{m}_{(ba)i} \\
&= \sum_{b \in \mathcal{N}_a \backslash \{c\}} \left( \sum_j \sum_l \sum_n \boldsymbol{W}_{ij}^{(2)} \boldsymbol{W}_{j(ln)}^{(1)} \boldsymbol{e}_{\text{CBF}}^{\text{rad}}(x_{ca})_{ln} \boldsymbol{e}_{\text{CBF}}^{\text{SH}}(\varphi_{bac})_l \right) \boldsymbol{m}_{(ba)i} \\
&= \sum_j \boldsymbol{W}_{ij}^{(2)} \sum_l \left( \sum_n \boldsymbol{W}_{j(ln)}^{(1)} \boldsymbol{e}_{\text{CBF}}^{\text{rad}}(x_{ca})_{ln} \right) \left( \sum_{b \in \mathcal{N}_a \backslash \{c\}} \boldsymbol{e}_{\text{CBF}}^{\text{SH}}(\varphi_{bac})_l \boldsymbol{m}_{(ba)i} \right).
\end{aligned} \tag{31}$$

For a bilinear layer we use

$$
\begin{aligned}
\boldsymbol{m}_{(ca)i} &= \sum_{b \in \mathcal{N}_a \setminus \{c\}} \left( \left( \boldsymbol{W}^{(1)} \boldsymbol{e}_{\mathrm{CBF}}(x_{ca}, \varphi_{bac}) \right)^T \mathbf{W}^{(2)} \boldsymbol{m}_{(ba)} \right)_i \\
&= \sum_{b \in \mathcal{N}_a \setminus \{c\}} \sum_{i'} \sum_{j} \left( \sum_{l} \sum_{n} \boldsymbol{W}_{j(ln)}^{(1)} \boldsymbol{e}_{\mathrm{CBF}}^{\mathrm{rad}}(x_{ca})_{ln} \boldsymbol{e}_{\mathrm{CBF}}^{\mathrm{SH}}(\varphi_{bac})_l \right) \mathbf{W}_{iji'}^{(2)} \boldsymbol{m}_{(ba)i'} \\
&= \sum_{j} \sum_{i'} \mathbf{W}_{iji'}^{(2)} \sum_{l} \left( \sum_{n} \boldsymbol{W}_{j(ln)}^{(1)} \boldsymbol{e}_{\mathrm{CBF}}^{\mathrm{rad}}(x_{ca})_{ln} \right) \left( \sum_{b \in \mathcal{N}_a \setminus \{c\}} \boldsymbol{e}_{\mathrm{CBF}}^{\mathrm{SH}}(\varphi_{bac})_l \boldsymbol{m}_{(ba)i'} \right).
\end{aligned}
\tag{32}
$$

Note that since $\boldsymbol{W}^{(1)}$ is shared across layers we only need to calculate the sum over $n$ once.

## E  Variance after message passing

The layer-wise variance after sum aggregation is

$$
\mathrm{Var}_i \Big[ \sum_{b \in \mathcal{N}_a} \boldsymbol{m}_{(ba)i} \Big] = \sum_{b \in \mathcal{N}_a} \mathrm{Var}_i[\boldsymbol{m}_{(ba)i}] + \sum_{b \in \mathcal{N}_a} \sum_{c \in \mathcal{N}_a \setminus \{b\}} \mathrm{Cov}_i[\boldsymbol{m}_{(ba)i}, \boldsymbol{m}_{(ca)i}].
\tag{33}
$$

This variance depends on the number of neighbors in $\mathcal{N}_a$. However, we consistently found that rescaling the output depending on $\mathcal{N}_a$ has negative effects on the accuracy. The likely reason for this is that atomic interactions scale roughly linearly with neighborhood size. Moreover, the covariance in Eq. (33) is not zero since all messages $\boldsymbol{m}_{ba}$ are transformed using the same weight matrices. We therefore best estimate this variance empirically.

For a Hadamard product-based message passing filter (and analogously for a bilinear layer) we have

$$
\mathrm{Var}_i[F_i \boldsymbol{m}_i] = \mathrm{Cov}_i[F_i^2, \boldsymbol{m}_i^2] + (\mathrm{Var}_i[F_i] + \mathbb{E}_i[F_i]^2)(\mathrm{Var}_i[\boldsymbol{m}_i] + \mathbb{E}_i[\boldsymbol{m}_i]^2) - (\mathrm{Cov}_i[F_i, \boldsymbol{m}_i] + \mathbb{E}_i[F_i]\mathbb{E}_i[\boldsymbol{m}_i])^2.
\tag{34}
$$

The main problem with this covariance is the non-zero quadratic covariance $\mathrm{Cov}_i[F_i^2, \boldsymbol{m}_i^2]$. We again estimate this variance empirically based on a data sample.

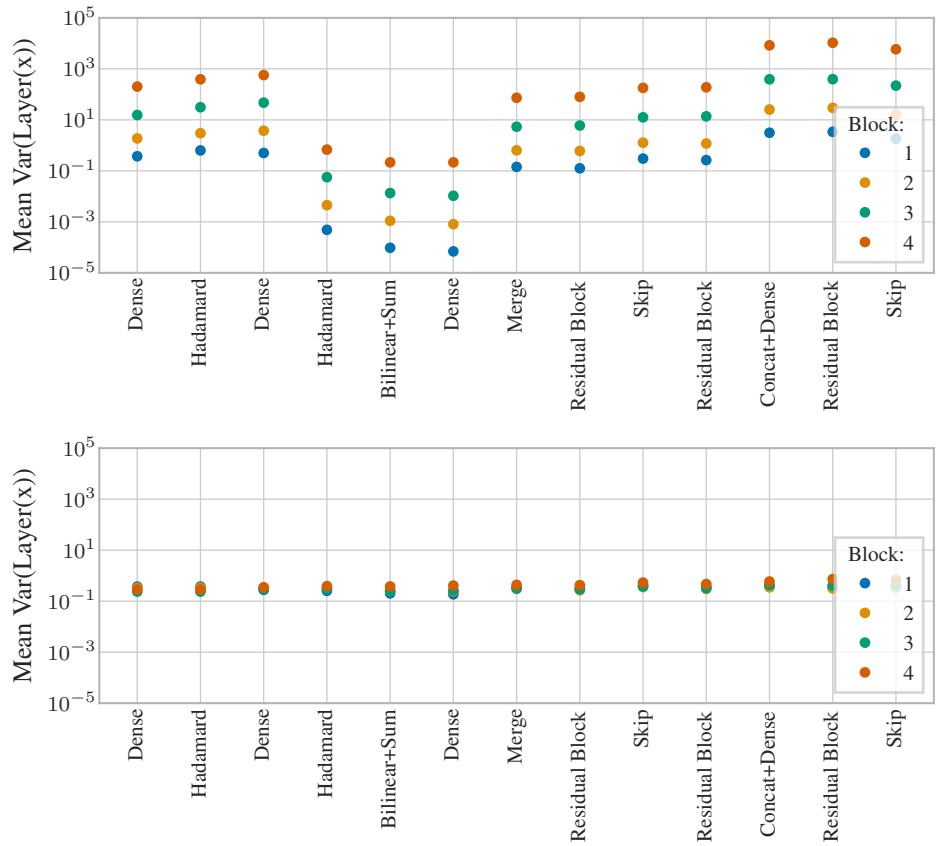

Figure 3: Layer-wise activation variance. GemNet's variance varies strongly between layers and increases significantly after each block without scaling factors (top). Introducing scaling factors successfully stabilizes the variance (bottom).

# F   GemNet architecture

We use 4 stacked interaction blocks and an embedding size of 128 throughout the model. For the basis functions we choose $N_{\text{SHBF}} = N_{\text{CHBF}} = 7$ and $N_{\text{SRBF}} = N_{\text{CRBF}} = N_{\text{RBF}} = 6$. For the weight tensor of the bilinear layer in the interaction block we use $N_{\text{bilinear,SBF}} = 32$ and $N_{\text{bilinear,CBF}} = 64$. We found that sharing the first weight matrix in Eq. (11), the down projection, resulted in the same validation loss but reduced the training time by up to 15 %. The down projection size was chosen as 16 for the radial and circular basis and 32 for the spherical basis.

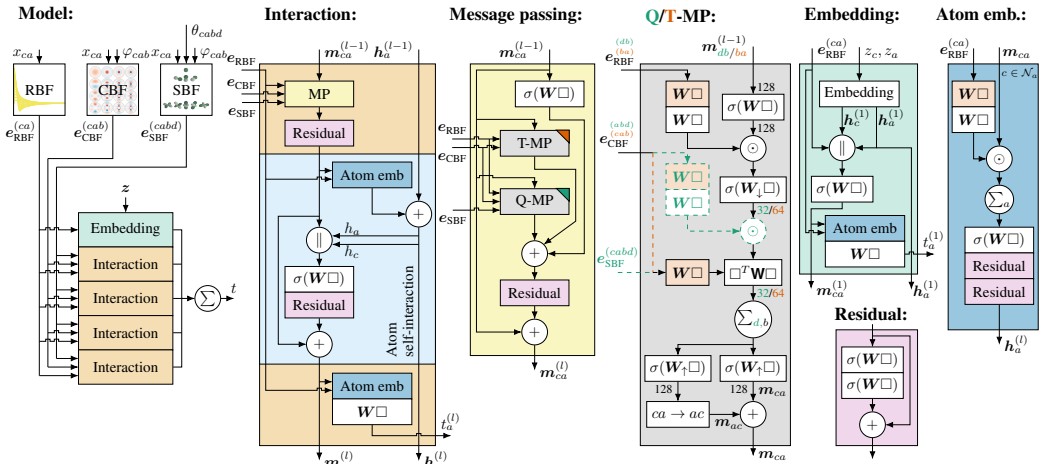

Figure 4: The full GemNet architecture. □ denotes the layer's input, ∥ concatenation, $\sigma$ a non-linearity (we use SiLU in this work [20]), and orange a layer with weights shared across interaction blocks. Differences between two-hop message passing (Q-MP) and one-hop message passing (T-MP) are denoted by dashed lines. Numbers next to connecting lines denote embedding sizes.

# G   Training and hyperparameters

Table 6: Model and training hyperparameters.

| Hyperparameters | | | |
|---|---|---|---|
| Interaction cutoff $c_{\text{int}}$ | | 10 Å | |
| Embedding cutoff $c_{\text{emb}}$ | | 5 Å | |
| Learning rate | | $1 \times 10^{-3}$ | |
| EMA decay | | 0.999 | |
| Weight decay | | $2 \times 10^{-6}$ | |
| Decay epochs | | 1200 | |
| Decay rate | | 0.01 | |
| Decay factor on plateau | | 0.5 | |
| Gradient clipping threshold | | 10.0 | |
| Envelope exponent | | 5 | |
| Force weighting factor $\rho$ | | 0.999 | |
| | MD17 | MD17@CCSD(T) | Coll |
| Train set size | 1000 | 950 | 120 000 |
| Val. set size | 1000 | 50 | 10 000 |
| Max epochs | 2000 | 2000 | 400 |
| Evaluation interval (epochs) | 10 | 10 | 2 |
| Decay on plateau patience (epochs) | 50 | 50 | 10 |
| Decay on plateau cooldown (epochs) | 50 | 50 | 10 |
| Warm-up epochs | 10 | 10 | 1 |
| Batch size | 1 | 1 | 32 |

We subtract the mean energy from each molecule in MD17 to obtain a training target similar to atomization energy. We train on eV for energies and eV Å$^{-1}$ for forces. As a training objective we use the weighted loss function

$$\mathcal{L}_{\text{MD}}(\boldsymbol{X}, \boldsymbol{z}) = (1 - \rho)\left| f_\theta(\boldsymbol{X}, \boldsymbol{z}) - \hat{t}(\boldsymbol{X}, \boldsymbol{z}) \right| + \frac{\rho}{N} \sum_{i=1}^{N} \sqrt{\sum_{\alpha=1}^{3} \left( -\frac{\partial f_\theta(\boldsymbol{X}, \boldsymbol{z})}{\partial \boldsymbol{x}_{i\alpha}} - \hat{F}_{i\alpha}(\boldsymbol{X}, \boldsymbol{z}) \right)^2 }, \tag{35}$$

with force weighting factor $\rho = 0.999$. We found the selection of the batch size to be of great influence on the model's performance for the MD17(@CCSD) dataset. Changing the batch size from

32 to 1 resulted in an approx. 25 % lower validation MAE. The learning rate of $1 \times 10^{-3}$ and the selection of the embedding cutoff $c_{emb} = 5\,\text{Å}$ and interaction cutoff $c_{int} = 10\,\text{Å}$ are rather important hyperparameters as well, see Table 8. We optimized the model using AMSGrad [50] with weight decay [40] in combination with a linear learning rate warm-up, exponential decay and decay on plateau. However, we did not apply the weight decay for the initial atom embeddings, biases and frequencies (used in the radial basis). Without weight decay the force MAE was around 3 % higher on COLL (not on OC20). Gradient clipping and early stopping on the validation loss were used as well. In addition, we divided the gradients of weights that are shared across multiple blocks by the number of blocks the weights are shared for, which resulted in a small gain in accuracy. The model weights for validation and test were obtained using an exponential moving average (EMA) with decay rate 0.999. The used hyperparameters can be found in Table 6. The combined model on revised MD17 was trained with a batch size of 10.

We used a slightly adapted model for the OC20 dataset. It uses 128 Gaussian radial basis functions instead of spherical Bessel functions, which do not depend on the degree $l$ of the spherical harmonic. We furthermore used only three interaction blocks, an atom and edge embedding size of 512, an embedding cutoff of $6\,\text{Å}$, a learning rate of $5 \times 10^{-4}$, no weight decay, only learning rate decay on plateau with a patience of 15 000 steps and a factor of 0.8 (no warm-up or exponential decay), and a batch size of 2048.

## H  Additional experimental results

Table 7: MAE for direct force predictions on MD17 in meV/Å. The increased speed of direct force predictions comes at a significant cost of accuracy. Note that the direct models are still more accurate than many previous models.

|  | GemNet-Q | GemNet-T | GemNet-dQ | GemNet-dT |
|---|---|---|---|---|
| Aspirin | 9.4 | 9.5 | 17.8 | 18.0 |
| Benzene[9] | 6.3 | 6.3 | 8.5 | 8.0 |
| Benzene[10] | 1.5 | 1.4 | 2.5 | 2.3 |
| Ethanol | 3.8 | 3.7 | 6.4 | 6.8 |
| Malonaldehyde | 6.9 | 6.7 | 11.5 | 12.5 |
| Naphthalene | 2.2 | 2.4 | 5.2 | 5.9 |
| Salicylic acid | 5.4 | 5.5 | 12.9 | 13.2 |
| Toluene | 2.6 | 2.6 | 6.1 | 5.7 |
| Uracil | 4.5 | 4.2 | 11.7 | 10.9 |

Table 8: Impact of the cutoff on force MAE on COLL. Results reported in meV/Å after 500 000 training steps. Increasing the interaction cutoff to $10\,\text{Å}$ slightly reduces the error. Decreasing the embedding cutoff to $3\,\text{Å}$ significantly increases the error.

| $c_{emb}/\text{Å}$ | $c_{int}/\text{Å}$ | MAE |
|---|---|---|
| 5 | 10 | 27.0 |
| 5 | 5 | 28.2 |
| 3 | 10 | 33.4 |
| 3 | 5 | 35.3 |

Table 9: Force MAE for MD17 in meV/Å. GemNet using 1000 training samples compared to SchNet using 50 000 samples. GemNet outperforms SchNet on six out of eight molecules – despite using 50x fewer samples.

|  | SchNet 50k | GemNet-Q | GemNet-T |
|---|---|---|---|
| Aspirin | 14.3 | **9.4** | 9.5 |
| Benzene[9] | 7.4 | **6.3** | **6.3** |
| Benzene[10] | - | 1.5 | **1.4** |
| Ethanol | **2.2** | 3.8 | 3.7 |
| Malonaldehyde | **3.5** | 6.9 | 6.7 |
| Naphthalene | 4.8 | **2.2** | 2.4 |
| Salicylic acid | 8.2 | **5.4** | 5.5 |
| Toluene | 3.9 | **2.6** | **2.6** |
| Uracil | 4.8 | 4.5 | **4.2** |

Table 10: Effect of adding our independent improvements to DimeNet$^{++}$ on force MAE for COLL in meV/Å. In this experiment we increased the basis embedding size of DimeNet$^{++}$ from 8 to 16 to eliminate this bottleneck. All improvements have a significant effect.

| Model | Forces |
|---|---|
| DimeNet$^{++}$ | 41.1 |
| with symmetric message passing | 37.5 |
| with bilinear layer | 38.6 |
| with scaling factors | 40.0 |

Table 11: Force MAE for the revised MD17 dataset [11] in meV/Å. On average, GemNet outperforms FCHL19 by 52 % and even UNiTE by 5 %, which is a $\Delta$-ML approach based on quantum mechanical features [49].

|  | FCHL19 | UNiTE | GemNet-Q | GemNet-T |
|---|---|---|---|---|
| Aspirin | 20.9 | **7.8** | 9.7 | 9.5 |
| Benzene | 2.6 | 0.7 | 0.7 | **0.5** |
| Ethanol | 6.2 | 4.2 | **3.6** | **3.6** |
| Malonaldehyde | 10.3 | 7.1 | 6.7 | **6.6** |
| Naphthalene | 6.5 | 2.4 | **1.9** | 2.1 |
| Salicylic acid | 9.5 | **4.1** | 5.3 | 5.5 |
| Toluene | 8.8 | 2.9 | 2.3 | **2.2** |
| Uracil | 4.2 | **3.8** | 4.1 | **3.8** |

Table 12: Force MAE of different models (number of parameters in parentheses) for the MD17 dataset in meV/Å. GemNet performs worse with an embedding size of 64, but still substantially better than previous models with more parameters.

|  | PaiNN (600k) | DimeNet (1.9M) | GemNet-T 64 (490k) | GemNet-T (1.9M) |
|---|---|---|---|---|
| Aspirin | 14.7 | 21.6 | 11.2 | **9.5** |
| Benzene[9] | - | 8.1 | - | **6.3** |
| Benzene[10] | - | - | **1.1** | 1.4 |
| Ethanol | 9.7 | 10.0 | 5.1 | **3.7** |
| Malonaldehyde | 14.9 | 16.6 | 7.8 | **6.7** |
| Naphthalene | 3.3 | 9.3 | 3.3 | **2.4** |
| Salicylic acid | 8.5 | 16.2 | 6.9 | **5.5** |
| Toluene | 4.1 | 9.4 | 3.3 | **2.6** |
| Uracil | 6.0 | 13.1 | 5.3 | **4.2** |

Table 13: Force MAE of GemNet on the revised MD17 dataset [11] in meV/Å when using individual models for each molecule ("Individual") versus a single model for all molecules ("Combined"). The combined setting is harder to learn, leading to a higher error in most cases. GemNet-Q performs better than GemNet-T in this setting.

|  | GemNet-Q | | GemNet-T | |
|---|---|---|---|---|
|  | Individual | Combined | Individual | Combined |
| Aspirin | 9.7 | 10.0 | 9.5 | 9.9 |
| Benzene | 0.7 | 0.5 | 0.5 | 0.6 |
| Ethanol | 3.6 | 4.4 | 3.6 | 4.9 |
| Malonaldehyde | 6.7 | 7.7 | 6.6 | 8.3 |
| Naphthalene | 1.9 | 1.9 | 2.1 | 2.2 |
| Salicylic acid | 5.3 | 4.6 | 5.5 | 5.0 |
| Toluene | 2.3 | 2.2 | 2.2 | 2.5 |
| Uracil | 4.1 | 4.1 | 3.8 | 4.3 |

## I   Computation time

The models were trained primarily using Nvidia GeForce GTX 1080Ti GPUs. For MD17 and MD17@CCSD training the direct force prediction variants took less than two days, GemNet-Q and GemNet-T took around 6 days per molecule but with very little progress after the 100 hour mark. However, thanks to the memory efficient implementation and the low batch size used, several models were trained in parallel on a single GPU. On the COLL dataset training the direct force prediction variants took around 24 hours each. GemNet-T trained for 60 hours, while GemNet-Q took 6 days. However, after 60 hours GemNet-Q is already within 5 % of its final validation error and outperforms GemNet-T by a large margin. Note that the training time reduces dramatically when using a larger batch size, at the cost of a slightly higher MAE on MD17.

Table 14: Runtime per batch of Toluene molecules on an Nvidia GeForce GTX 1080Ti in seconds. GemNet-T is comparably fast to previous methods. Note that NequIP requires roughly 10x more training epochs than GemNet for convergence [4]. Using direct force predictions and only one-hop message passing significantly accelerates training and inference (GemNet-dT). Efficient aggregation allows for the usage of a bilinear layer instead of a Hadamard product at no additional cost (GemNet-Q vs. Hadamard-Eff) and enables training with higher batch sizes (Hadamard-Eff vs. Hadamard-NonEff). Note that our implementation does not focus on runtime and can likely be significantly optimized.

| | batch size 32 | | batch size 4 | |
|---|---|---|---|---|
| | Training | Inference | Training | Inference |
| DimeNet$^{++}$ | 0.357 | 0.065 | 0.283 | 0.031 |
| NequIP (l=1) | 0.066 | 0.042 | 0.070 | 0.044 |
| NequIP (l=3, reflections) | 0.336 | 0.206 | 0.327 | 0.197 |
| GemNet-Q | 1.067 | 0.376 | 0.628 | 0.099 |
| GemNet-T | 0.397 | 0.088 | 0.299 | 0.038 |
| GemNet-dQ | 0.369 | 0.264 | 0.106 | 0.052 |
| GemNet-dT | 0.134 | 0.067 | 0.065 | 0.020 |
| Hadamard-Eff | 1.077 | 0.392 | 0.632 | 0.103 |
| Hadamard-NonEff | OOM | 0.378 | 0.633 | 0.103 |