# OpenReview forum: "GemNet: Universal Directional Graph Neural Networks for Molecules"
_NeurIPS.cc/2021/Conference — NeurIPS 2021 Poster_

### Official Review · Reviewer_Uuo9 · 2021-07-02

**Rating:** 6
**Confidence:** 4

**Summary:**

The paper proposes GemNet, an extension to DimeNet++ for predicting molecular properties. The additions include two-hop message passing, symmetric message-passing and efficient bilinear layers. GemNet achieves state of the art results on the MD17 and COLL benchmarks.

**Limitations And Societal Impact:**

Clarify practical implications of universality proof (size of cutoff etc., see above).

**Main Review:**

The paper is well motivated and structured. Some aspects could be explained in more detail, in particular the "symmetric message-passing" scheme in Sec 5. The experiments and ablation studies demonstrate the performance and careful design choices of GemNet.

The sections on the universality of spherical  and directional message-passing seem a bit misleading to me. If I'm not mistaken, the proofs are only valid for general functions $F_\text{sphere}$ / $F_\text{TFN}$. However, in practice one needs to introduce a cutoff. In this case, the "Picasso problem" mentioned in the introduction becomes an issue again, if one only uses rotationally-invariant representations. Thus, the central issue is not just the universality of the architecture in theory, but the tradeoff between rotational symmetries of the representation (angular momentum of spherical harmonics) and locality. A discussion on this and how this might be connected to the two-hop scheme would be interesting.

Another minor issue is the discussion of inference time and model size:  the authors state that those are "on par" with previous models. While this is true for DimeNet, it is not the case for NequIP and PaiNN, which are much faster and smaller, while outperforming DimeNet. The timing of Toluene in Table 9 for NequIP (100ms, bs 4) also seems a bit high compared to the original paper (16ms, bs 1). It would be interesting to see also the accuracy and performance of a scaled-down GemNet with a comparable number of parameters.

**Time Spent Reviewing:**

3

---

> ### Author Response · Authors · 2021-08-09
> **Full expressivity with cutoff, new runtime measurements, scaled-down GemNet**
>
> Thank you for your review and for highlighting that our paper is well motivated and structured! We have revised our **explanation of symmetric message passing**, as suggested:
> > Whenever we have a directional embedding $m_{ca}$, we also have the opposing embedding $m_{ac}$, since both are based on the same cutoff $c_{\text{emb}}$. Whether we associate the embedding $m_{ca}$ or $m_{ac}$ with atom $a$ is arbitrary. A more principled approach is to \emph{jointly} interpret both embeddings as a representation of the atom pair $a$ and $c$. In this view, an update to $m_{ca}$ should also influence $m_{ac}$. Normally, this would require executing the above message passing scheme twice, once for updating $m_{ca}$ based on $m_{db}$, and once for updating $m_{ac}$ based on $m_{db}$. We propose to circumvent this double execution by calculating the update (\cref{eq:core_geom}) only once and then using it for both $m_{ca}$ and $m_{ac}$. To preserve the distinction between the two directions and ensure that $m_{ca} \neq m_{ac}$, we then transform the two embeddings using two separate learnable weight matrices. One single message passing update thus carries information for both embeddings, which is then dissected by the two weight matrices. In practice, this only requires a simple re-indexing operation that maps the edge $ca$ to $ac$.
>
> We also extended the explanations in our theoretical section, as per reviewer YFgs's comments. We are happy to improve any other section that is difficult to follow.
>
> ### Universality and cutoffs
>
> Cutoffs are indeed an interesting aspect that has so far been ignored in all of the theoretical literature, since it is very easy to construct a counterexample for a model with cutoff: Just have 2 points and a long-range interaction. As soon as the points move outside the cutoff, the model will predict a constant value, regardless of the actual positions. Analyzing models with cutoffs thus requires carefully defining the function space, which is highly non-trivial --- especially considering that the electromagnetic interaction actually is long ranged and thus requires special treatment. We have to leave this for future work.
>
> However, our proof is based on showing the **equivalence** between spherical models and SO(3)-based models. Even with a cutoff, spherical models remain **as expressive as SO(3)-based models**. Our statement that SO(3) representations are not needed thus extends to the case with cutoffs, and is not misleading. Note that spherical representations are equivariant, not invariant. The Picasso problem thus does not apply here. By interpreting edge embeddings as a form of spherical representation, we show that they also capture the full relative directional information (if we perform two-hop message passing) --- thus also circumventing the Picasso problem.
>
> Moreover, we can combine our Theorem 2 with a recently published theoretical result to prove that our method of directly predicting forces is even universal for predicting equivariant properties \[Villar, Hogg, Storey-Fisher, Yao, Blum-Smith. June 2021. Scalars are universal: Gauge-equivariant machine learning, structured like classical physics\].
>
> ### New runtime measurements
>
> Since the official NequIP code has been published in the meantime and we found an error in our GemNet time measurements, we have now updated the runtimes in the appendix. Note that the time in the NequIP paper was measured on a 32-core CPU, not an Nvidia GTX 1080Ti as in our paper.
>
> | Model                     | Train, BS32 | Inf, BS32   | Train, BS4 | Inf, BS4|
> |---------------------------|-------------|-------------|------------|---------|
> | DimeNet$^{++}$            | 0.357       | 0.065       | 0.283      | 0.031   |
> | NequIP (l=1)              | 0.066       | 0.042       | 0.070      | 0.044   |
> | NequIP (l=3, reflections) | 0.336       | 0.206       | 0.327      | 0.197   |
> | GemNet-Q                  | 1.067       | 0.376       | 0.628      | 0.099   |
> | GemNet-T                  | 0.397       | 0.088       | 0.299      | 0.038   |
>
> ### Scaled-down GemNet
>
> As stated in lines 317-318, GemNet uses as many parameters as DimeNet (1.9M), while PaiNN uses 600k. As suggested, we have trained a small GemNet with half the embedding size (64, using 490k parameters). The following table shows that this small GemNet still performs significantly better than PaiNN. Note that the GemNet-T results have improved slightly since submission due to a reduced batch size.
>
> |                |  PaiNN (600k) | GemNet-T 64 (490k) | GemNet-T (1.9M) |
> |----------------|--------|-------------|--------------|
> | Aspirin        |  14.7  |    11.2     |      9.5     |
> | Benzene        |    -   |     1.1     |      1.4     |
> | Ethanol        |   9.7  |     5.1     |      3.7     |
> | Malonaldehyde  |  14.9  |     7.8     |      6.7     |
> | Naphthalene    |   3.3  |     3.3     |      2.4     |
> | Salicylic acid |   8.5  |     6.9     |      5.5     |
> | Toluene        |   4.1  |     3.3     |      2.6     |
> | Uracil         |   6.0  |     5.3     |      4.2     |

---

### Official Review · Reviewer_LzdH · 2021-07-15

**Rating:** 7
**Confidence:** 3

**Summary:**

The authors propose a model Geometric Message Passing Neural Network (GemNet) which leverages directed edge embeddings and a new two-hop message passing scheme. The model is more expressive and performs convincingly better than the most related baseline (DimeNet) as well as other baselines across a large range of datasets.

**Limitations And Societal Impact:**

See main review.

**Main Review:**

The model contributions seem theoretically sound (subject to my limited expertise on the topic) and in a way the model feels like a natural extension to DimeNet to include torsion angles. The paper is well written, with sufficient information for reproducibility, and the results compared to previous baselines (and specially compared to DimeNet) are compelling. While I did not go into the details, I appreciate the proof of universality of representations for rotationally invariant models, which is great way to qualify the expressivity of the model. Finally, the problem it tackles has very direct applications in science problems, and is of very high interest in state of the art research in rotationally invariant models.

I think my main criticism is about the comparison between GemNet-Q and GemNet-T. Authors say “This suggests that one-hop message passing is expressive enough for some practical use cases, but two-hop message passing gives an advantage for the more challenging task of fitting multiple molecules at once”. As a potential user, it would be great to have a bit more insight (e.g. some rule of thumb) of when to use one or the other. Perhaps some investigation using synthetic datasets would have helped conveying this. For example considering that GemNet-Q, is technically more expressive than GemNet-T, in the cases where GemNet-Q performs worse, is this due to overfitting, or due to optimization problems?

--------------

Post rebuttal:

Thank you for your responses and additional experiments and clarification on the comparison between GemNet-Q and GemNet-T. I would like to maintain my original rating, which was already very positive.


**Time Spent Reviewing:**

2

---

> ### Author Response · Authors · 2021-08-09
> **Clarity on GemNet-T vs. GemNet-Q**
>
> We are happy to hear that you find our paper to be theoretically sound, well written, that our results are compelling, and that the problem we tackle has very direct applications in science and is of very high interest in state of the art research in rotationally invariant models.
>
> The fact that GemNet-T performed better on MD17 in the submitted version of the paper is most likely due to overfitting. We have recently run an experiment with a reduced batch size on MD17, which improves performance overall and leads to GemNet-Q performing on par with GemNet-T. Thus, GemNet-Q now always performs on par or better than GemNet-T. This also means that the **comparison between GemNet-Q vs. T is clear now**, so we added this statement after line 311:
> > In general, we recommend using GemNet-Q when model accuracy is most important. When runtime and memory consumption are more of a concern, we recommend using GemNet-T.

---

### Official Review · Reviewer_Nzov · 2021-07-16

**Rating:** 7
**Confidence:** 2

**Summary:**

Many GNN models have recently been proposed for the problem of predicting molecular dynamics. These GNNs rely on rotationally invariant predictions since the underlying physics of the problem does not change with rotations. While such models have obtained good performance, there is limited theoretical understanding of these methods.

This paper makes two contributions to this task: first, it addresses the theoretical limitations by proving that GNNs with directed edge embeddings and two-hop message passing are universal approximators; second, it presents a new model called GemNet (an extension of Dimenet++ that uses two hop message passing), that obtains excellent results on COLL and MD17 datasets.

**Limitations And Societal Impact:**

The authors include a discussion of the limitations of their method.

**Main Review:**

## Originality
The theoretical contributions in the paper appear novel and interesting. The GemNet model builds upon the Dimenet and Dimenet++ models, but introduces some novel ideas, particularly the use of dihedral angles.

## Quality
The paper appears technically sound. The theoretical claims made by the authors have been proven rigorously and the new model is shown to outperform previous methods experimentally on MD17 and COLL datasets. The new model is a natural extension to the theoretical claims, which provides a good backing for the theory.

Stabilizing activation variance and efficient interactions that the authors use in the GemNet paper are orthogonal to the other improvements and they can also be used in Dimenet++. It would be good to include ablations that incorporate these improvements into Dimenet++ so we can more clearly see the benefits of including dihedral angles.

## Clarity
The paper was difficult to follow and suffers from mathiness. I would recommend the authors to try and simplify the presentation.

## Significance
The methods presented in the paper are likely to be useful for many molecular chemistry problems.

**Time Spent Reviewing:**

4

---

> ### Author Response · Authors · 2021-08-09
> **Improved readability and further ablation studies**
>
> We are happy to hear that you find our theoretical contributions novel and interesting, technically sound and rigorously proven; and that our model provides good backing to the theory.
>
> ### More ablation studies
> We agree that GemNet includes multiple further improvements that are of independent interest, and have therefore investigated these in the ablation study in Table 4 and the runtimes in Appendix H. Additionally, we have now performed the suggested ablation studies with DimeNet++. These are the resulting force MAEs:
> - baseline: 41.1meV
> - activation scaling: 40.0meV
> - efficient bilinear layer: 38.6meV
> - symmetric message passing: 37.5meV
>
> All 3 changes thus yield improvements on DimeNet++ as well as GemNet. Note that we had to increase the basis embedding size to 16 for these results. This seems to be an information bottleneck in regular DimeNet++. The baseline includes this change.
>
> ### Improved readability
>
> While we understand that the mathematical treatment in Sections 3 & 4 requires some knowledge of group theory, we believe that this treatment is essential for rigorously proving the universality of spherical models (Theorem 2) and subsequently of two-hop message passing with directed edge embeddings. These statements would not be accurate without the necessary mathematical backing.
>
> Note that we have made several improvements in response to reviewer YFgs's comments. We are more than happy to improve the paper further if you point us to the parts that are difficult to follow.
>
> **[Update]**: We've updated the above ablation study and added a discussion.
> **[Update 2]**: We've updated the above ablation study after we found that it was necessary to increase the basis embedding size in DimeNet++ to properly leverage our improvements.

---

> ### Author Response · Authors · 2021-08-27
> **Updated ablation studies**
>
> As promised, we have updated the DimeNet++ ablation studies after the corresponding experiments had finished. The changes achieved a test force MAE of:
>
> - baseline: 41.1meV
> - activation scaling: 40.0meV
> - efficient bilinear layer: 38.6meV
> - symmetric message passing: 37.5meV
>
> **All 3 proposed orthogonal improvements thus yield benefits for DimeNet++ as well as GemNet**. Note that we had to increase the basis embedding size to 16 to fully leverage the bilinear layer and symmetric message passing. This seems to be an information bottleneck in regular DimeNet++. The baseline includes this change.

---

### Official Review · Reviewer_YFgs · 2021-07-17

**Rating:** 5
**Confidence:** 4

**Summary:**

* The paper proposes to use a type of networks that can be shown to be universal approximators for functions that are permutation equivariant, rotationally and translationally invariant.
* This property is required to model functions that work on molecular structures. In essence, this provides a very strong prior on the functions we want to learn.
* The experiments show that by enforcing this structure learning on molecular graphs can be improved (higher quality/label efficiency) compared to previous work. While, the experimental section is quite compact and heavily compressed nothing stood out as obviously wrong but I might be missing a detail.

Overall I like the high level idea behind the paper and the results seem good. While, at a high level I am able to understand the paper, I am really struggling to understand the details and I do not feel like I would be able to reproduce the results or understand the nuances of the architecture. I am aware that the code is released, but this should not be a replacement for a clearly written paper. For this reason I suggest a score of marginally below the acceptance threshold. In the main review I have highlighted the issues and points were I got stuck.



**Ethical Concerns:**

Nothing obvious.

**Limitations And Societal Impact:**

Clarity

**Main Review:**


## Section 3
The tensor field network definition is a notational mess and I am struggling to properly understand the actual structure/meaning.
* Eq. 1: K and D are not defined K(D) properly, $K,D\inN$ is provided but not what they stand for.
* Eq. 2: what does $l_o$ and $m_o$ stand for?
* Eq. 2: indexing of the clebsch-gordan coefficients is not clear to me. (And a wikipedia search does not help either since it has a different notation). I digged deeper in the references in this paper but unfortunately the TFN paper refers to Griffiths (Introduction to quantum mechanics) which is 147 dollars on amazon at the day of the review and a bit to expensive/cumbersome to order and read for this review. Given that many people at Neurips might be in a similar situation I would recommend the authors to actually explain this properly.
* Eq. 3. $F_{sphere}=\sum_{l,m}$ what is l and m? Is R^l a part of the learned polynomial function.  Can you better describe how I should think about Y.  Looking at this is seems similar to a fourier basis but for functions defined on the sphere? It is not immediately obvious how to think about the function on the sphere. Eq. 3 depends on a direction $\hat{r}$ and $x_b-x_a$, which is then reduced into a direction and a magnitude component. Providing an intuition for how to think about such a function would be helpful for the general ML researcher to better understand this paper.

## Section 4
* Eq. 5 what is $F_2$? What is $r_o$ and $r_i$.

## Section 5:
* There is a definition of $r_o$ and $r_i$. This would have been helpful in the previous section.
* Figure 1, right hand side is not helpful. The flat arrow going in a circle does not really mean much to me. Why is a and a,b darker?.

## Figure 2
* Residual is not defined. are all the W's the same? What is the atom emb. What is W_arrowup, what is W_arrowdown. What is z. How is the first mca initialized?




**Time Spent Reviewing:**

5-6 possibly more.

---

> ### Author Response · Authors · 2021-08-09
> **Improved notation and explanations**
>
> Thank you for spending such a considerable amount of time on reviewing our paper, diving into group theory and the SO(3) group, and for providing some invaluable feedback!
>
> We realize that group theory is a difficult topic to get into. Unfortunately, a conference paper does not provide the required space for a proper introduction into these topics --- especially considering that this part of the paper is primarily a summary of previous work. As such, this part is not relevant to reproducing our experimental results.
>
> We have made several changes, which address all of the mentioned issues and make the paper **more approachable**:
> - Pointed the reader to some concise introductions:
>  > Note that this section is not intended as an introduction to the SO(3) group. For a concise introduction in the context of machine learning see e.g. Weiler et al. [53] or Kondor et al. [34].
> - added more explanations for K and D,
> > where $D \in \mathbb{N}$ denotes the function's maximum polynomial degree, $K(D) \in \mathbb{N}$ is chosen such that \cref{th:univ_tfn} is fulfilled (\citet{dym_universality_2021} prove the existence of this function without specifying it)...
> - for the degree $l_o$ and order $m_o$,
> > We index the output degree and order with $l_o$ and $m_o$, the learned filter with $l_f$ and $m_f$, and the input with $l_i$ and $m_i$.
> - for the Clebsch-Gordan coefficients (note that their exact values are not relevant to our discussion and not used in our model),
> > The Clebsch-Gordan coefficients $C_{(l_f, m_f), (l_i, m_i)}^{(l_o, m_o)}$ arise from decomposing the tensor product of two input SO(3) representations (the filter and input representations) into a sum of output representations. Their exact values are not relevant for this discussion.
> - for the spherical harmonics $Y_{lm}$ and $Y^{(l)}_m$, as used in $F_{sphere}$,
> > ...the real spherical harmonics $Y_{lm}$ with degree $l$ and order $m$. The spherical harmonics are the basis for the Fourier transformation of functions on the sphere, analogously to sine waves for functions on $\mathbb{R}$.
> - for the filter $F_2$,
> > To incorporate this we add a convolution with a learned filter $F_2$, which can only improve the model's expressiveness.
> - and for the directions $r_i$ and $r_o$ (the directional mesh is described in lines 151-161)
> > ...where $R_b$ denotes the directional mesh of atom $b$ with mesh directions denoted by $\hat{r}_i$, and $\hat{r}_o$ specifies the output direction.
> - We updated Fig. 1, removing the grey color, noting that the circle arrow denotes rotation, and changing the caption to:
> > Angles used in geometric message passing. The dihedral angle $\theta_{cabd}$ becomes visible when rotating the molecule so that atoms $a$ and $b$ lie on top of each other (right).
> - As noted in the text, Fig. 2 does not describe all low-level layers. Please see the appendix for a comprehensive version that includes the Residual, embedding, and atom embedding layers. We now point the reader to that version in the figure caption as well. We decided to use this simplified version in the main paper so we don't overwhelm the reader with all the details. As common in model figures, each $W$ denotes a separate learnable weight matrix. $W_\uparrow$ and $W_\downarrow$ just hint at changes in embedding size (up and down projections).
> - We now fully describe the model's inputs and outputs in Section 6:
> > GemNet predicts the molecular energy $E$ and forces $F \in \mathbb{R}^{3 \times n}$ based on the atomic positions $X \in \mathbb{R}^{3 \times n}$ and the atomic numbers $z \in \mathbb{N}^n$.
>
> These changes should significantly improve the readability and accessibility, especially of the theoretical section. If you find further issues we are more than happy to further improve the paper!

---

### Author Response · Authors · 2021-08-09
**Clearer writing, further experiments, and a stronger theoretical result**

We would like to thank the reviewers for their diligent effort and hope that our responses help clarify any last doubts. Based on your invaluable feedback we were able to significantly improve the paper's clarity.

We have also independently run experiment on two other datasets in the last months and would like to highlight the additional results we have obtained:
- On the QM7-X dataset we substantially outperform previous models, which achieve a force MAE of 39.9meV/A. GemNet-T achieves 5.1meV/A and GemNet-Q 4.3meV/A, outperforming the previous state of the art by an extreme 9x. This further demonstrates the capabilities of GemNet when moving beyond single molecules (as in MD17).
- We furthermore evaluated the large GemNet-T on the open catalyst (OC20) dataset, where we currently achieve a validation force MAE of 0.0215, outperforming previous models by 20%. Similar to previous results \[Hu, Shuaibi, Das, Goyal, Sriram, Leskovec, Parikh, Zitnick. 2021. ForceNet: A Graph Neural Network for Large-Scale Quantum Calculations\], we found that direct force predictions consistently perform better on this dataset. This furthermore makes our model significantly faster and more memory efficient than previous state-of-the-art models on OC20.

We would furthermore like to highlight a recent theoretical result, published in Villar, Hogg, Storey-Fisher, Yao, Blum-Smith. June 2021. "Scalars are universal: Gauge-equivariant machine learning, structured like classical physics". If we combine their result with our Theorem 2, it is easy to show that the method we propose for direct force predictions is actually universal. Therefore, two-hop message passing with directed edges is even a **universal approximator for rotationally equivariant predictions**. If the reviewers agree to this, we would like to include this stronger result in our camera-ready paper.

---

### Decision · Program_Chairs · 2021-09-27

**Decision:**

Accept (Poster)

**Comment:**

This paper was generally well received. Reviewers unanimously agreed that the paper was interesting and impactful, especially given the widespread interest in message passing neural networks to model atomistic systems. It seems clear that NeurIPS is an appropriate venue for this publication.

The authors provided a number of clarifications and further experiments in response to the reviews that helped to address several concerns (in particular regarding baselines against existing models such as NequIP and ablation studies). The main issues that could not be addressed during the rebuttal period were the clarity of exposition which several reviewers commented on. I would encourage the authors to work on simplifying the paper as much as possible in the time leading up to the camera ready.